# GRF: Learning a General Radiance Field for 3D Scene Representation and Rendering

## Abstract

We present a simple yet powerful implicit neural function that can represent and render arbitrarily complex 3D scenes in a single network only from 2D observations. The function models 3D scenes as a general radiance field, which takes a set of 2D images with camera poses and intrinsics as input, constructs an internal representation for each 3D point of the scene, and renders the corresponding appearance and geometry of any 3D point viewing from an arbitrary angle. The key to our approach is to explicitly integrate the principle of multi-view geometry to obtain the internal representations from observed 2D views, such that the learned implicit representations empirically remain multi-view consistent. In addition, we introduce an effective neural module to learn general features for each pixel in 2D images, allowing the constructed internal 3D representations to be general as well. Extensive experiments demonstrate the superiority of our approach.

## 1 Introduction

Understanding the precise 3D structure of a real-world environment and realistically re-rendering it from free viewpoints is a key enabler for many critical tasks, ranging from robotic manipulation and navigation to augmented reality. Classic approaches to recover the 3D scene geometry mainly include the structure from motion (SfM) (Ozyesil et al., 2017) and simultaneous localization and mapping (SLAM) (Cadena et al., 2016) pipelines. However, they are limited to reconstruct sparse and discrete 3D point clouds which are unable to contain geometric details.

The recent advances in deep neural networks have yielded rapid progress in 3D modeling and understanding. Most of them focus on the explicit 3D shape representations such as voxel grids (Choy et al., 2016), point clouds (Fan et al., 2017), or triangle meshes (Wang et al., 2018). However, these representations are discrete and sparse, limiting the recovered 3D structures to extremely low spatial resolution. In addition, these networks usually require large-scale 3D shapes for supervision, resulting in the trained models being over-fitting particular datasets and unable to generalize to novel scenes. In fact, it is also costly and even infeasible to collect high-quality 3D labels.

Encoding geometries into multilayer perceptrons (MLPs) (Mescheder et al., 2019; Park et al., 2019) recently emerges as a promising direction in 3D reconstruction and understanding from 2D images. Its key advantage is the ability to model 3D structures in a continuous way instead of discrete, and therefore it has the potential to achieve unlimited spatial resolution in theory. However, most methods of this pipeline focus on individual objects. In addition, many of them require 3D geometry for supervision to learn the 3D shapes from images. By introducing a recurrent neural network based renderer, SRNs (Sitzmann et al., 2019) is among the early work to learn implicit surface representations only from 2D images, but it fails to capture complicated scene geometries and renders over-smoothed images. Alternatively, by leveraging the volume rendering to synthesize new views for 2D supervision, the very recent NeRF (Mildenhall et al., 2020) directly encodes the radiance fields of complex 3D scenes within the weights of MLPs, achieving an unprecedented level of fidelity for challenging 3D scenes. Nevertheless, it has two major limitations: 1) since each 3D scene is encoded into all the weights of MLPs, the trained network (*i.e.,* a learned radiance field) can only represent a single scene, and is unable to generalize across novel scenarios; 2) because the shape and appearance of each spatial 3D location along a light ray is only optimized by the available pixel RGBs, the learned implicit representations of that location are lack of the general geometric patterns, resulting in the synthesized images being less photo-realistic.

In this paper, we propose a **g**eneral **r**adiance **f**ield (GRF), a simple yet powerful implicit function that can represent and render complex 3D scenes where there may be multiple objects with cluttered background. Our GRF takes a set of 2D images with camera poses and intrinsics, a 3D query point and its query viewpoint (i.e., the camera location $xyz$) as input, and predicts the RGB value and volumetric density of that query point. Basically, this neural function learns to represent a 3D scene from sparse 2D observations, and infers the shape and appearance of that scene from previously unobserved viewing angles. Note that, the inferred shape and appearance of any particular 3D query point explicitly take into account its local geometric patterns from the available 2D observations. In particular, the proposed GRF consists of four components:

- Extracting the general 2D visual features for every light ray from the input 2D observations;
- Reprojecting the corresponding 2D features back to the query 3D point using the principle of multi-view geometry;
- Selecting and aggregating all the reprojected features for the query 3D point, while the visual occlusions are implicitly considered;
- Rendering the aggregated features of the query 3D point along a particular query viewpoint, and producing the corresponding RGB and volumetric density.

These four components enable our GRF to be significantly different from all existing 3D scene representation approaches. 1) Compared with the classic SfM/SLAM systems, our GRF can represent the 3D scene structure with smooth and continuous surfaces. 2) Compared with the neural approaches based on explicit voxel grids, point clouds and meshes, our GRF learns continuous 3D representations without requiring 3D data for training. 3) Compared with the existing implicit representation methods such as SDF (Park et al., 2019), SRNs (Sitzmann et al., 2019) and NeRF (Mildenhall et al., 2020), our GRF can represent arbitrarily complicated 3D scenes and has remarkable generalization to novel scenarios. In addition, the learned 3D representations carefully consider the general geometric patterns for every 3D spatial location, allowing the rendered views to be exceptionally realistic with fine-grained details. Our key contributions are:

- We propose a general radiance field to implicitly represent the 3D scene structure and appearance purely from 2D images. It has remarkable generalization to novel scenes in a single forward pass.
- We explicitly integrate the principle of multi-view geometry to learn geometric details for each 3D query point along every query light ray. This allows the synthesized 2D views to be superior.
- We demonstrate significant improvement over baselines on three large-scale datasets and provide intuition behind our design choices through extensive ablation studies.

## 2 RELATED WORK

**Classic Multi-view Geometry.** Classic approaches to reconstruct 3D geometry from images mainly include SfM and SLAM systems, which firstly extract and match hand-crafted geometric local features and then apply bundle adjustment for both shape and camera motion estimation (Hartley & Zisserman, 2004). Although they can recover visually satisfactory 3D models, the reconstructed shapes are usually sparse, discrete point clouds. In contrast, our GRF learns an implicit function to represent the continuous 3D structures from images. Notably, however, the principle of classic multi-view geometry is explicitly integrated into our GRF to learn accurate and general features for every spatial location.

**Geometric Deep Learning.** The recent advances of deep neural nets have yielded impressive progress in recovering explicit 3D shapes from either single or multiple images, including the voxel grid (Choy et al., 2016; Yang et al., 2019), octree (Riegler et al., 2017; Christian et al., 2017), point cloud (Fan et al., 2017; Qi et al., 2017) and triangle mesh (Wang et al., 2018; Groueix et al., 2018; Nash et al., 2020) based approaches. However, most of these methods only focus on individual 3D objects, whilst only few pipelines (Song et al., 2017; Tulsiani et al., 2018; Gkioxari et al., 2019) attempt to learn the structures for complex 3D scenes. Although these neural networks can predict realistic 3D structures of objects and scenes, they have two limitations. First, almost all of them require ground truth 3D labels to supervise the networks, resulting in the learned representations being unable to generalize to novel real-world scenes. Second, since the recovered 3D shapes are discrete, they are unable to preserve high-resolution geometric details. Being quite different, our GRF learns

to infer continuous 3D shapes only from a set of 2D images with camera poses and intrinsics, which can be cheaply acquired and also allow better generalization across real-world scenes.

**Neural Implicit 3D Representations.** The implicit representation of 3D shapes recently emerges as a promising direction to recover 3D geometries. It is initially formulated as level sets by optimizing neural nets which map $xyz$ locations to an occupancy field (Mescheder et al., 2019; Saito et al., 2019) or a distance function (Park et al., 2019). The subsequent work (Niemeyer et al., 2020; Sitzmann et al., 2019; Liu et al., 2019b; Atzmon & Lipman, 2020a;b) introduce differentiable rendering functions, allowing 2D images or raw 3D point clouds to supervise the networks, but only producing decent results on individual objects. Using neural radiance fields instead, the latest NeRF (Mildenhall et al., 2020) and the succeeding NSVF (Liu et al., 2020), NeRF-wild (Martin-Brualla et al., 2020), GRAF (Schwarz et al., 2020) demonstrate impressive results to represent complex 3D scenes. However, they do not take into account the local geometric patterns for spatial locations, thereby the rendered 2D images being less realistic and sub-optimal. Additionally, both NeRF and NeRF-wild can only represent a single scene, and are unable to generalize to novel scenarios. Uniquely, our GRF maps any set of images to the corresponding 3D scene structure with geometric details.

**Novel View Synthesis and Neural Rendering.** Novel view synthesis involves generating unseen views of a scene captured by multiple images. Existing methods usually learn an embedding of a scene and then estimate a new image given a viewing angle, including GAN based methods (Goodfellow et al., 2014; Radford et al., 2016), variational auto-encoders (Kingma & Welling, 2014), autoregressive models (Oord et al., 2016), and other generative frameworks (Eslami et al., 2018). Although photo-realistic single images can be generated, these methods tend to learn the manifold of 2D images, instead of exploiting the underlying 3D geometry for consistent multi-view synthesis.

Neural rendering techniques (Fried et al., 2020; Kato et al., 2020) have recently been actively investigated and integrated into 3D reconstruction pipelines, where there are no ground truth 3D data available but only 2D images for supervision. To render the discrete voxel grids (Yan et al., 2016; Rezende et al., 2016; Tulsiani et al., 2017), point clouds (Insafutdinov & Dosovitskiy, 2018), meshes (Kato et al., 2018; Chen et al., 2019; Liu et al., 2019a), and implicit surfaces (Liu et al., 2019b; Remelli et al., 2020), most of these techniques are designed with differentiable and approximate functions, but sacrifice the sharpness of synthesized images and the accuracy of estimated 3D shapes. By contrast, our GRF leverages the existing volume rendering method (Mildenhall et al., 2020) which is naturally differentiable and can accurately render the RGB per light ray.

## 3 GRF

### 3.1 OVERVIEW

Our GRF models the complex 3D scenes including the geometry and appearance as a neural function $f_\mathbf{W}$ where $\mathbf{W}$ represent learnable parameters. This function takes a set of $M$ images together with their camera poses and intrinsics $\{(\mathcal{I}_1, \boldsymbol{\xi}_1, \boldsymbol{K}_1) \cdots (\mathcal{I}_m, \boldsymbol{\xi}_m, \boldsymbol{K}_m) \cdots (\mathcal{I}_M, \boldsymbol{\xi}_M, \boldsymbol{K}_M)\}$, which are sampled from a complex 3D scene, a query 3D point $p = \{x_p, y_p, z_p\}$, and its query viewpoint $\mathcal{V}_p = \{x_p^v, y_p^v, z_p^v\}$ as input, and then predicts the RGB value $\{r_p, g_p, b_p\}$ and the volumetric density $d_p$ of that query 3D point $p$ observed from the query viewpoint $\mathcal{V}_p$. Formally, it is defined as below:

$$(r_p, g_p, b_p, d_p) = f_\mathbf{W}\Big( \Big\{(\mathcal{I}_1, \boldsymbol{\xi}_1, \boldsymbol{K}_1) \cdots (\mathcal{I}_M, \boldsymbol{\xi}_M, \boldsymbol{K}_M)\Big\}, \{x_p, y_p, z_p\}, \{x_p^v, y_p^v, z_p^v\} \Big) \quad (1)$$

Basically, this function is a general radiance field (GRF) which parameterizes arbitrary 3D scenes observed by the input $M$ images. In the meantime, it returns both the appearance and geometry, when being queried at any location $p$ from any viewpoint $\mathcal{V}$ in 3D space.

As illustrated in Figure 1, the proposed GRF firstly extracts general features for each light ray hitting through every pixel, and then reprojects those features back to the query 3D point $p$. After that, the corresponding RGB value and volumetric density of $p$ are inferred from those features given a specific query viewpoint. This simple design of GRF follows the principle of classic multi-view geometry (Hartley & Zisserman, 2004), therefore driving the learned implicit representations to be multi-view consistent. In particular, for any query 3D point in space, its features are always the same when being viewing from different angles. Hence, its rendered RGBs and volumetric densities

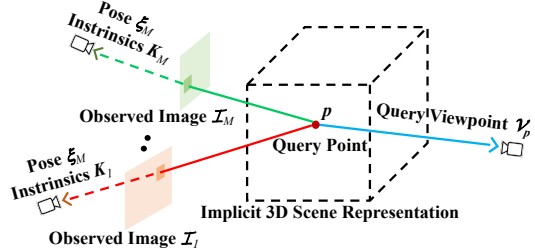

Figure 1: Overview of the proposed general radiance field.

are more likely to be consistent from different viewpoints. To realize our neural function $f_{\mathbf{W}}$, we introduce four components: 1) A feature extractor for every 2D pixel; 2) A reprojector to transform 2D features to 3D space; 3) An aggregator to obtain general features for a 3D point; and 4) A neural renderer to infer the appearance and geometry for that 3D point. All these components are connected and trained end-to-end with arbitrary scenes.

## 3.2 EXTRACTING GENERAL FEATURES FOR 2D PIXELS

The whole input set of observed 2D images together describes the geometry and appearance of a 3D scene, while each pixel of those images describes a specific point of that scene. This module aims to extract the general features of each pixel, so to learn the regional description and geometric patterns for each light ray. A naive approach is to directly use the raw *rgb* values as the pixel features. However, this is sub-optimal because the raw *rgb* values are sensitive to lighting conditions, environmental noise, etc. In order to learn more general and robust patterns for

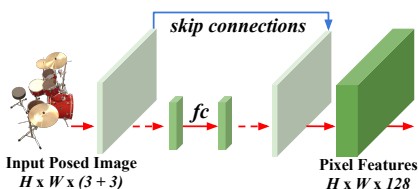

Figure 2: The CNN module to extract pixel features.

each pixel, we turn to use a more powerful encoder-decoder based convolutional neural network (CNN). As shown in Figure 2, our CNN module is designed with the follow two features:

- Instead of directly feeding raw RGB images into the CNN module, we stack (duplicate) the corresponding viewpoint, i.e., the $xyz$ location of the camera, to each pixel of the image. This allows the learned pixel features to be explicitly aware of its relative position in the 3D scene space. Note that, we empirically find that stacking the additional camera rotation and intrinsics to each pixel does not noticeably improve the performance, especially for scenes centered at the origin.

- We use skip connections between the encoder and decoder to preserve high frequency local features for each pixel, while optionally integrating a couple of fully connected (fc) layers in the middle of the CNN module to learn global features. The mixture of hierarchical features tends to to be more general and representative, effectively aiding the network in practice.

Details of the CNN module is presented in the appendix and all input images share the same CNN module. Note that, there are many ways to construct a pixel feature extractor, but identifying an optimal CNN module here is out the scope of this paper.

## 3.3 REPROJECTING 2D FEATURES TO 3D SPACE

Considering the extracted pixel features are a compact description of the light ray emitting from the camera center up to the 3D scene surface, we naturally reproject the pixel features back to the 3D space along the light ray. Since there are no depth scans paired with RGB images, it is impossible to determine which particular 3D surface point the pixel features belong to. In this module, we preliminarily regard the pixel features as the representation of every location along the light ray in 3D space. With this simple formulation, every 3D point can theoretically have a copy of its corresponding 2D pixel features from each 2D image. Formally, given a 3D point $p$, an observed 2D view $\mathcal{I}_m$ together with the camera pose $\boldsymbol{\xi}_m$ and the intrinsics $\boldsymbol{K}_m$, the corresponding 2D pixel features $\mathbf{F}_p^m$ is retrieved by the reprojection operation below:

$$\mathbf{F}_p^m = \mathcal{P}\Big(\{\mathcal{I}_k, \boldsymbol{\xi}_m, \boldsymbol{K}_m\}, \{x_p, y_p, z_p\}, \mathbf{I}_m\Big) \tag{2}$$

*where* the function $\mathcal{P}()$ follows the principle of multi-view geometry (Hartley & Zisserman, 2004) and $\mathbf{I}_m$ represents the image features extracted by the CNN module in Section 3.2. However, since the pixels of 2D images are discrete and bounded within a certain spatial size, while the 3D points are continuous in the scene space, after the 3D point $p$ is projected to the plane of image $\mathcal{I}_m$, we apply two approximations to deal with the following issues.

- If the point lies inside of the image, we simply select the nearest pixel and duplicate its features to the 3D point. Note that, more advanced techniques may be applied to address the discretization issue, such as linear interpolation or designing a kernel function.
- If the point lies outside of the image, we assign a zero vector to the 3D point, which means there is no information observed. In fact, we empirically find that the nearest interpretation can also achieve good performance, but it is only applicable for relatively small-scale scenes.

Overall, the above simple reprojection operation explicitly retains the extracted 2D pixel features back to 3D space via the principle of geometry, and the module is naturally differentiable.

### 3.4    OBTAINING GENERAL FEATURES FOR 3D POINTS

Given a query 3D point $p$, it is able to retrieve a feature vector from each input image. However, given a set of input images, it is challenging to obtain a final feature vector for the point $p$, because:

- The total number of input images for each 3D scene is variable and there is no order for images. As a consequence, the retrieved feature vectors are also unordered and have arbitrary size.
- Since there are no depth scans paired with the input RGB images, it is unable to decide which feature vectors are the true descriptions of the query point due to visual occlusions. Ideally, these features can be aware of the relative distance to the query point and then selected automatically.

To tackle these critical issues, we formulate this problem as an attention aggregation process. In particular, given the query 3D point $p$, its query viewpoint $\mathcal{V}_p$, and the set of retrieved pixel features $\{\mathbf{F}_p^1 \cdots \mathbf{F}_p^m \cdots \mathbf{F}_p^M\}$:

- For each retrieved feature vector $\mathbf{F}_p^m$, we firstly use shared MLPs to integrate the information of query point $p$, generating a new feature vector $\hat{\mathbf{F}}_p^m$ which is aware of the relative distance to the query point $p$. Formally, it is defined as:

$$\hat{\mathbf{F}}_p^m = MLPs(\mathbf{F}_p^m \oplus [x_p, y_p, z_p]), \quad (\oplus \text{ means concatenation in neural nets}) \tag{3}$$

- After obtaining the new set of position-aware features $\{\hat{\mathbf{F}}_p^1 \cdots \hat{\mathbf{F}}_p^m \cdots \hat{\mathbf{F}}_p^M\}$, we use the existing attention aggregation methods such as AttSets (Yang et al., 2020) and Slot Attention (Locatello et al., 2020) to compute a unique feature vector $\bar{\mathbf{F}}_p$ for the query 3D point $p$. Basically, the attention mechanism learns a unique weight for all input features and then aggregates them together. According to the theoretical analysis in (Yang et al., 2020) and (Locatello et al., 2020), the selected attention mechanisms are permutation invariant with regard to the input set of feature vectors and can process an arbitrary number of elements. Formally, it is defined as:

$$\bar{\mathbf{F}}_p = \mathcal{A}\big(\hat{\mathbf{F}}_p^1 \cdots \hat{\mathbf{F}}_p^m \cdots \hat{\mathbf{F}}_p^M\big), \quad (\mathcal{A} \text{ is an existing attention function}) \tag{4}$$

Above all, for every query 3D point $p$, its final features $\bar{\mathbf{F}}_p$ explicitly preserve the general geometric patterns retrieved from the input 2D observations and are also made aware of the particular 3D point location within the scene space. This allows the 3D point features $\bar{\mathbf{F}}_p$ to be general and representative for its own geometry and appearance.

### 3.5    RENDERING 3D FEATURES

With the learned features $\bar{\mathbf{F}}_p$ for any query 3D point $p$, this module aims to infer its volumetric density $d_p$ in 3D space and the corresponding RGB values $\{r_p, g_p, b_p\}$ given the query viewpoint $\mathcal{V}_p$. In particular, we exactly follow the simple MLP-based design of NeRF (Mildenhall et al., 2020) in the rendering process. As illustrated in Equation 5, the volumetric density $d_p$ is modeled as a function of the point features $\bar{\mathbf{F}}_p$ only. This allows the estimated density $d_p$ at the query point $p$ to remain unchanged under different query viewpoints, thus maintaining consistent geometry for each point. The RGB is modeled as a function of both point features and the query viewpoint. Details of

these MLPs refers to NeRF (Mildenhall et al., 2020).

$$d_p = MLPs(\bar{\mathbf{F}}_p), \qquad (r_p, g_p, b_p) = MLPs\Big(\bar{\mathbf{F}}_p, \{x_p^v, y_p^v, z_p^v\}\Big) \qquad (5)$$

To render the color of any query ray passing through the 3D scene, the classic volume rendering (Kajiya & Herzen, 1984) is applied and the integral is estimated by sampling a discrete set of 3D points along the query ray. In particular, we exactly follow the formulation of ray marching and the hierarchical sampling strategy in NeRF (Mildenhall et al., 2020). Eventually, novel 2D images can be directly synthesized from our GRF by querying light rays. This allows the entire network to be trainable only with a set of 2D images, without requiring 3D data.

## 3.6 IMPLEMENTATION

The proposed four modules are connected and trained end-to-end. In the implementation, details of the CNN module and the attention module for different experiments are presented in the appendix. In the neural rendering module, all the designs of neural layers and other settings simply follow NeRF (Mildenhall et al., 2020). In our network, all the 3D locations and RGB values are processed by the positional encoding proposed by NeRF (Mildenhall et al., 2020). The L2 loss between the rendered RGB and the ground truth is used to optimize the whole neural network. All these details are presented in the appendix.

## 4 EVALUATION

### 4.1 EXPERIMENTS ON SHAPENETV2 DATASET

Following the experimental settings of SRNs (Sitzmann et al., 2019), we firstly evaluate our GRF on the chair and car classes of ShapeNetv2. Particularly, the chair has 4612 objects for training, 662 for validation and 1317 for testing, while the car has 2151 objects for training, 352 for validation and 704 for testing. Each training object has randomly sampled 50 images with a resolution of $128 \times 128$ pixels. We conduct the following three groups of experiments.

- Group 1: Novel-view synthesis of seen objects in the training split of the same category. In particular, we train two separate models on the training split of chairs and cars, and these two models are separately tested on the trained objects to generate completely novel views. During testing, the model is fed with 50 trained views and infers another 251 novel views for evaluation.

- Group 2: Novel-view synthesis of unseen objects in the testing split of the same category. The trained two models in Group 1 are tested on novel objects of the same category. During testing, the model is fed with 2 novel views of each novel object, inferring 251 novel views for evaluation.

- Group 3: Similar to Group 2, but only 1 novel view is fed into the model for novel view synthesis.

Basically, the experiments of Group 1, 2 and 3 are exactly the same as in SRNs (Sitzmann et al., 2019) which is also the state of the art on this ShapeNetv2 dataset. Table 1 compares the quantitative results of our GRF and four baselines. Note that, the recent NeRF and NSVF are not scalable to learn large number of scenes simultaneously because each scene is encoded into the network parameters.

**Analysis.** 1) Our method outperforms SRNs on 50-view reconstruction, demonstrating the strong capability of our GRF for 3D scene representation. Figure 3 shows the qualitative results. 2) Our GRF lags behind SRNs for novel scene reconstruction in both Group 2 and 3. Fundamentally, this is because our GRF solves a much harder problem than SRNs. In particular, our network directly infers the novel scene representation in a single forward pass, while SRNs needs to be retrained on all novel scenes to optimize the latent code. As a result, the experiments of Group 2 and 3 are significantly in favour of SRNs. 3) To further demonstrate the advantage of GRF over SRNs, we directly evaluate the trained SRNs model on unseen objects (of the same category) without retraining. For comparison, we also directly evaluate the trained GRF model on the same novel objects. Figure 4 shows the qualitative results. It can be seen that if not retrained, SRNs completely fails to reconstruct unseen car instances, but randomly generates similar cars from learned prior knowledge, primarily because its latent code has not been updated from the unseen objects. In contrast, by learning pixel local patterns, our GRF generalizes well to novel objects directly. This highlights the superiority of the simple formulation of our GRF.

Table 1: Comparison of the PSNR (in dB) and SSIM of reconstructed images in the ShapeNetv2 dataset by our GRF, the deterministic GQN (Eslami et al., 2018), TCO (Tatarchenko et al., 2015), WRL (Worrall et al., 2017) and SRNs (Sitzmann et al., 2019). The higher the scores, the better the synthesized novel views. **Note that, our GRF solves a harder problem than SRNs, because GRF infers the novel scene representation in a single forward pass, while SRNs cannot.**

| | 50 Images (Group 1) | | 2 Images (Group 2) | | 1 Image (Group 3) | |
|---|---|---|---|---|---|---|
| | Chairs | Cars | Chairs | Cars | Chairs | Cars |
| TCO | 24.31 / 0.92 | 20.38 / 0.83 | 21.33 / 0.88 | 18.41 / 0.80 | 21.27 / 0.88 | 18.15 / 0.79 |
| WRL | 24.57 / 0.93 | 19.16 / 0.82 | 22.28 / 0.90 | 17.20 / 0.78 | 22.11 / 0.90 | 16.89 / 0.77 |
| dGQN | 22.72 / 0.90 | 19.61 / 0.81 | 22.36 / 0.89 | 18.79 / 0.79 | 21.59 / 0.87 | 18.19 / 0.78 |
| SRNs | 26.23 / **0.95** | 26.32 / **0.94** | **24.48 / 0.92** | **22.94 / 0.88** | **22.89 / 0.91** | **20.72 / 0.85** |
| **GRF (Ours)** | **27.10** / 0.95 | **27.21** / 0.94 | 22.65 / 0.88 | 22.06 / 0.86 | 21.25 / 0.86 | 19.84 / 0.81 |

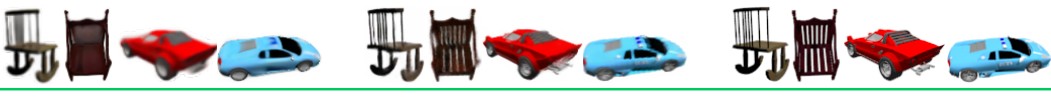

SRNs         GRF (Ours)         Ground Truth

Figure 3: Qualitative results of SRNs and our GRF for 50-view reconstruction of chairs and cars.

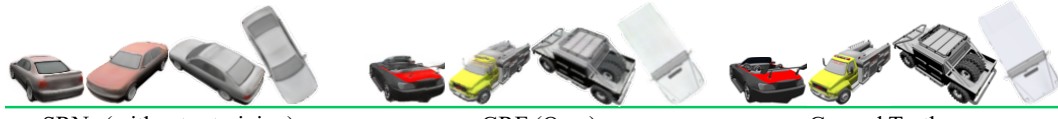

SRNs (without retraining)       GRF (Ours)       Ground Truth

Figure 4: Qualitative results of SRNs (without retraining) and our GRF for 50-view reconstruction of unseen cars. SRNs fails to recover faithful shapes for unseen cars without retraining.

## 4.2 EXPERIMENTS ON SYNTHETIC-NERF DATASET

We further evaluate our approach on the more complex dataset Synthetic-NeRF (Mildenhall et al., 2020). It consists of pathtraced images of 8 synthetic scenes with much more complicated geometry and realistic non-Lambertian materials. Each scene has 100 views for training and 200 novel views for testing. Each image has a high resolution of $800 \times 800$ pixels.

**Single-scene Learning.** To demonstrate the representational capability of our GRF, we train a single model for each synthetic scene, following the same experimental settings of NSVF (Liu et al., 2020). Table 2 compares the quantitative results. Our method is better or on par with the state-of-the-art NSVF on all metrics. Figure 6 shows the qualitative results.

Table 2: The average scores of PSNR, SSIM and LPIPS (Zhang et al., 2018) for our GRF, SRNs (Sitzmann et al., 2019), NV (Lombardi et al., 2019), LLFF (Mildenhall et al., 2019), NeRF (Mildenhall et al., 2020) and NSVF (Liu et al., 2020) on the Synthetic-NeRF dataset for single-scene learning.

| | PSNR↑ | SSIM↑ | LPIPS↓ |
|---|---|---|---|
| SRNs | 22.26 | 0.846 | 0.170 |
| NV | 26.05 | 0.893 | 0.160 |
| LLFF | 24.88 | 0.911 | 0.114 |
| NeRF | 31.01 | 0.947 | 0.081 |
| NSVF | 31.74 | 0.953 | **0.047** |
| **GRF(Ours)** | **32.06** | **0.960** | 0.048 |

**Novel-scene Generalization.** To evaluate the generalization of GRF on complex scenes, we train a single model simultaneously from randomly selected 4 scenes of Synthetic-NeRF, i.e., Chair, Mic, Ship, and Hotdog, and then conduct the following two groups of experiments.

- Group 1: The trained model is directly tested on the remaining 4 novel scenes of Synthetic-NeRF dataset, i.e., Drums, Lego, Materials, and Ficus. Fundamentally, this experiment is to evaluate whether the learned limited features can truly generalize to new scenarios. This is extremely challenging because there are only 4 training scenes and the overall shapes of the 4 novel scenes are dramatically different from the trained ones.

- Group 2: The trained model is further finetuned on each of the four novel scenes with 100, 1k, and 10k iterations separately. In total, we obtain $4 \times 3 = 12$ new models. For comparison, we also train NeRF on each of the four novel scenes with 100, 1k, 10k iterations from scratch. This group of experiments evaluates how the initially-learned general features of our GRF improve novel scene learning.

**Analysis.** Table 3 compares the quantitative results. We can see that: 1) In Group 1, our GRF can indeed generalize to novel scenes with complex and completely different overall geometries, demonstrating the effectiveness of learning point local features in GRF. As shown in Figure 5, the overall shape and appearance of Lego can be satisfactorily recovered, though it has never been seen before. 2) In Group 2, our GRF can quickly learn high-quality scene representations given a sparse number of training pixels, thanks to the initially learned general features. Compared with the NeRF trained from scratch, the learned GRF significantly improves novel scene learning, achieving much better results given the same amount of training signals of new scenes. Figure 5 qualitatively shows the generalization of GRF and more results are presented in appendix.

Table 3: The average scores of PSNR, SSIM and LPIPS for GRF and NeRF on four novel scenes of Synthetic-NeRF in Group 1&2.

|  | PSNR↑ | SSIM↑ | LPIPS↓ |
|---|---|---|---|
| **GRF (Group 1)** | 13.62 | 0.763 | 0.246 |
| NeRF (Group 2, 100 iters) | 15.15 | 0.752 | 0.359 |
| NeRF (Group 2, 1k iters) | 19.81 | 0.809 | 0.228 |
| NeRF (Group 2, 10k iters) | 23.35 | 0.875 | 0.137 |
| **GRF (Group 2, 100 iters)** | 19.69 | 0.835 | 0.169 |
| **GRF (Group 2, 1k iters)** | 22.00 | 0.876 | 0.128 |
| **GRF (Group 2, 10k iters)** | 25.10 | 0.916 | 0.089 |

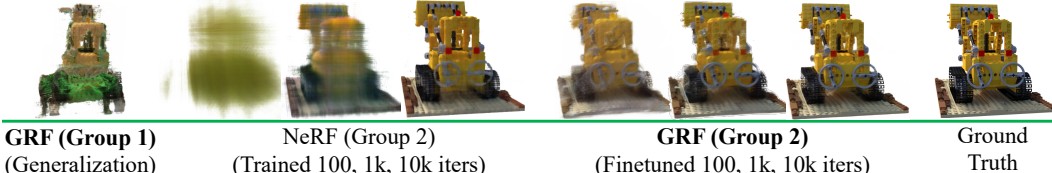

| GRF (Group 1) | NeRF (Group 2) | GRF (Group 2) | Ground |
|---|---|---|---|
| (Generalization) | (Trained 100, 1k, 10k iters) | (Finetuned 100, 1k, 10k iters) | Truth |

Figure 5: Qualitative results of our GRF for novel scene generalization.

## 4.3 EXPERIMENTS ON REAL-WORLD COMPLEX SCENES

We evaluate the novel view synthesis on complex real-world scenes captured by cellphones. There are 8 scenes, 5 from LLFF (Mildenhall et al., 2019) and 3 from NeRF. Each scene has 20 to 62 images, 1/8 of which for testing. All images have high resolution of $1008 \times 756$ pixels.

**Single-scene Learning.** We train a single model for each real-world scene, following the same experimental settings of NeRF. Table 4 compares the quantitative results. Note that, the recent NSVF (Liu et al., 2020) is unable to process these forward-facing scenes because the predefined voxels cannot represent the unbounded 3D space.

Table 4: Comparison of the average PSNR, SSIM and LPIPS scores of our GRF, SRNs, LLFF and NeRF in the challenging real-world dataset for single-scene learning.

|  | PSNR↑ | SSIM↑ | LPIPS↓ |
|---|---|---|---|
| SRNs | 22.84 | 0.668 | 0.378 |
| LLFF | 24.13 | 0.798 | 0.212 |
| NeRF | 26.50 | 0.811 | 0.250 |
| **GRF(Ours)** | **26.64** | **0.837** | **0.178** |

**Analysis.** Our method surpasses the state of the art by large margins, especially over the SSIM and LPIPS metrics. Compared with PSNR which only measures the average per-pixel accuracy, the metrics SSIM and LPIPS favor high quality of photorealism, highlighting the superiority of our GRF to generate truly realistic images. Figure 6 shows the qualitative results. As highlighted by the red circles, our GRF can generate fine-grained and natural geometries in pixel level, while NeRF produces many artifacts. This clearly demonstrates our GRF indeed learns general and precise pixel features from the observed 2D images for 3D scene representation and rendering.

Compared with Synthetic-NeRF, this dataset is much harder because of the complicated geometries and the inaccuracy of estimated camera poses. Therefore, it is non-trivial to conduct real-world experiments for novel-scene generalization or large-scale scene reconstruction on ScanNet (Dai et al., 2017) or "Tanks and Temples" (Knapitsch et al., 2017). We leave these for future exploration.

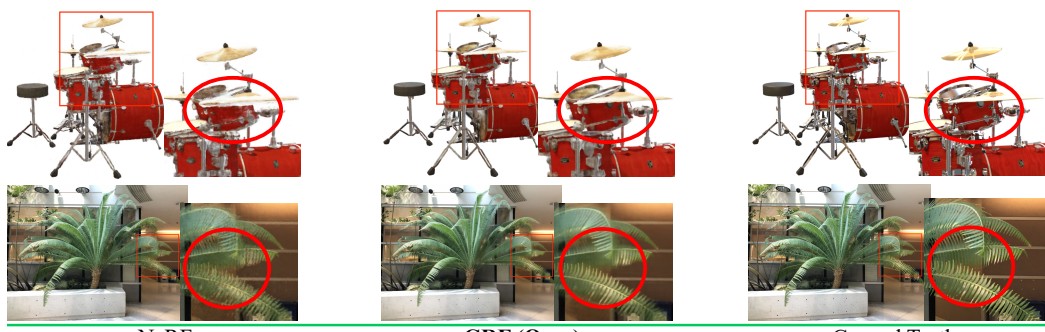

| NeRF | **GRF (Ours)** | Ground Truth |

Figure 6: Qualitative results on Synthetic-NeRF (1st row) and real-world datasets (2nd row).

## 4.4 ABLATION STUDY

To evaluate the effectiveness of the key components of our GRF, we conduct 3 groups of ablation experiments on the car category in ShapeNetv2 dataset. In particular, we train on the entire training split then randomly select 500 objects from the training split for testing. During testing, the model infers 50 novel views for each of these 500 objects for evaluation. Basically, this setting is similar to the experiment of Group 1 in Section 4.1.

Table 5: The average scores of PSNR and SSIM for ablated GRF in the subset of car category.

|  | PSNR↑ | SSIM↑ |
|---|---|---|
| (1) Remove Input Viewpoints | 20.13 | 0.807 |
| (2) Remove Pixel Local Features | 20.23 | 0.818 |
| (3) Replace Attention by Maxpool | 24.88 | 0.914 |
| **(4) The Full Model** | **27.16** | **0.942** |

• Group 1: The viewpoints of the input images are removed from the CNN module. The CNN module is not explicitly aware of the relative position of the pixel features.

• Group 2: The decoder of the CNN module is removed and each input image is encoded as a global feature vector. For any 3D query point which is rightly projected into the image boundary, that global feature vector is retrieved and reprojected to the query point. Fundamentally, this modified CNN module can be regarded as a hyper-network that learns a conditional embedding from input images and then feed it into NeRF, but it sacrifices the precise pixel local features.

• Group 3: The advanced attention module is replaced by max-pooling to aggregate the pixel features, aiming to investigate how the visual occlusion can be better addressed using soft attention.

• Group 4: The full model is trained and tested with the same settings for comparison.

**Analysis.** Table 5 compares the performance of all ablated models. We can see that: 1) The greatest impact is caused by the removal of image viewpoints from the CNN module and the lack of local pixel features to represent 3D points. It highlights that obtaining the position-aware and precise pixel features is crucial to represent 3D scenes from 2D images, while learning a simple hyper-network for NeRF cannot achieve comparable performance. 2) Using max-pooling to select the reprojected pixel features is sub-optimal to address the visual occlusions from different viewing angles.

To further investigate the attention module, the maximal attention scores are retrieved and visualized (detailed in appendix). It shows that the attention module can truly drive the network to focus on the visible pixel local features among the multiple intersected light rays. We also conduct additional ablation experiments (detailed in appendix), demonstrating that our GRF can indeed generalize well to novel scenes with severe visual occlusions and is robust to a variable number of input images.

## 5 CONCLUSION

The proposed method models complex 3D scenes as a general radiance field. We demonstrated that it can learn general and robust 3D point features from a set of 2D observations. By using the principle of multi-view geometry to precisely map 2D pixel features back to 3D space and leveraging the attention mechanism to implicitly address the visual occlusions, our GRF can synthesize truly realistic 2D novel views. However, there are still limitations that lead to the future work. (1) More advanced CNN modules can be designed to learn better pixel features. (2) Depth scans can be integrated into the network to explicitly address visual occlusions.

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

# A  APPENDIX

## A.1  DETAILS OF NETWORK ARCHITECTURE

Table 6: The CNN Module for Experiments on ShapeNetv2 Dataset.

| Type | Size/Channels | Activation | Stride |
|---|---|---|---|
| Input: embedding of RGB and viewpoint | - | - | - |
| L1: Conv $7 \times 7$ | 64 | ReLU | 2 |
| L2: Conv $3 \times 3$ | 128 | ReLU | 2 |
| L3: Conv $3 \times 3$ | 256 | ReLU | 2 |
| L4: Conv $3 \times 3$ | 512 | ReLU | 2 |
| L5: Conv $4 \times 4$ | 128 | ReLU | 4 |
| L6: Flatten / Tile | - | - | - |
| L7: Concat (L6, L4) | - | - | - |
| L8: Dilated Conv $3 \times 3$ | 256 | ReLU | 4 |
| L8: Concat (L8, L3) | - | - | - |
| L9: Dilated Conv $3 \times 3$ | 128 | ReLU | 8 |
| L9: Concat (L9, L2) | - | - | - |
| L10: Dilated Conv $3 \times 3$ | 64 | ReLU | 16 |
| L10: Concat (L10, L1) | - | - | - |
| L11: Dilated Conv $3 \times 3$ | 128 | ReLU | 32 |

Table 7: The CNN Module for Experiments on Synthetic-NeRF Dataset. The average pooling is added to aggressively downsample the feature maps.

| Type | Size/Channels | Activation | Stride |
|---|---|---|---|
| Input: embedding of RGB and viewpoint | - | - | - |
| L1: Conv $7 \times 7$ | 64 | ReLU | 2 |
| L2: Conv $3 \times 3$ | 128 | ReLU | 2 |
| L3: Conv $3 \times 3$ | 256 | ReLU | 2 |
| L4: Conv $3 \times 3$ | 512 | ReLU | 2 |
| L4: AveragePooling $5 \times 5$ | - | - | - |
| L5: Conv $5 \times 5$ | 128 | ReLU | 5 |
| L6: Flatten / Tile | - | - | - |
| L7: Concat (L6, L4) | - | - | - |
| L8: Deconv $3 \times 3$ | 256 | ReLU | 2 |
| L8: Concat (L8, L3) | - | - | - |
| L9: Deonv $3 \times 3$ | 128 | ReLU | 2 |
| L9: Concat (L9, L2) | - | - | - |
| L10: Deconv $3 \times 3$ | 64 | ReLU | 2 |
| L10: Concat (L10, L1) | - | - | - |
| L11: Deconv $3 \times 3$ | 128 | ReLU | 2 |

Table 8: The CNN Module for Experiments on the real-world Dataset. The average pooling is added to aggressively downsample the feature maps.

| Type | Size/Channels | Activation | Stride |
|------|---------------|------------|--------|
| Input: embedding of RGB and viewpoint | - | - | - |
| L1: Conv $7 \times 7$ | 64 | ReLU | 2 |
| L2: Conv $3 \times 3$ | 128 | ReLU | 2 |
| L3: Conv $3 \times 3$ | 256 | ReLU | 2 |
| L4: Conv $3 \times 3$ | 512 | ReLU | 2 |
| L4: AveragePooling $8 \times 8$ | - | - | - |
| L5: Conv $4 \times 4$ | 128 | ReLU | 4 |
| L6: Flatten / Tile | - | - | - |
| L7: Concat (L6, L4) | - | - | - |
| L8: Deconv $3 \times 3$ | 256 | ReLU | 2 |
| L8: Concat (L8, L3) | - | - | - |
| L9: Deconv $3 \times 3$ | 128 | ReLU | 2 |
| L9: Concat (L9, L2) | - | - | - |
| L10: Deconv $3 \times 3$ | 64 | ReLU | 2 |
| L10: Concat (L10, L1) | - | - | - |
| L11: Deconv $3 \times 3$ | 128 | ReLU | 2 |

Table 9: The Attention Module, AttSets (Yang et al., 2020), for Experiments on the ShapeNetv2 Dataset. The simple AttSets is computationally efficient and we choose it to train the large-scale ShapeNetv2 dataset.

| Type | Size/Channels | Activation |
|------|---------------|------------|
| Input: Concat($K \times 128$, embedding of viewpoint) | - | - |
| L1: fc | 256 | ReLU |
| L2: fc | 256 | ReLU |
| L3: fc | 256 | ReLU |
| L4: fc | 512 | ReLU |
| L5: fc | 512 | ReLU |
| L6: softmax(L5) | - | - |
| L7: sum(L6*L5, axis=-2) | - | - |
| L8: fc | 512 | ReLU |

We use Slot Attention as the pixel feature aggregation module for experiments on the Synthetic-NeRF and real-world datasets. In particular, we use two slots, two iterations, and the hidden size is 128. The final output two slots are flattened and a 256 dimensional vector is obtained. For details of Slot Attention refer to the paper (Locatello et al., 2020).

Details of the neural rendering layers and the volume rendering can be found in NeRF (Mildenhall et al., 2020). We set the positional embedding length $L = 5$ for all inputs to the CNN module, except the rotation, which we convert to quaternion and embed at $L = 4$.

During training, we feed the models between 2 and 6 views of each scene at each gradient step. We set the learning rate for the ShapeNetv2 models at 1e-4. We set the learning rate for leaves and orchids in the real-world dataset at 7e-5, and for the rest, we use 1e-4. For Synthetic-NeRF dataset, we use a learning rate of 1e-4. We use the Adam optimizer for all models, and train for 200k-300k iterations. At each gradient step, we take 1000 rays for ShapeNetv2 with 32 coarse samples and 64 fine samples, and 800 rays for the real-world and Synthetic-NeRF datasets with 64 coarse samples and 192 fine samples. We train each model on a single Nvidia-V100 GPU with 32GB VRAM.

During testing for the ShapeNetv2 reconstruction from 50 images, we feed the model the 5 closest views to the desired novel view by cosine similarity from the 50 training images. During testing for single-scene reconstruction, we feed the model the 4 closest views by cosine similarity to the desired novel view.

## A.2 DETAILS OF EXPERIMENTAL RESULTS ON THE SYNTHETIC-NERF DATASET

Table 10: Comparison of the PSNR (in dB), SSIM and LPIPS (Zhang et al., 2018) scores of our GRF, SRNs (Sitzmann et al., 2019), NV (Lombardi et al., 2019), NeRF (Mildenhall et al., 2020) and NSVF (Liu et al., 2020) in the Synthetic-NeRF dataset for single-scene learning.

| | Chair | Drums | Lego | Mic | Materials | Ship | Hotdog | Ficus | *Mean* |
|---|---|---|---|---|---|---|---|---|---|
| | | | | PSNR↑ | | | | | |
| SRNs | 26.96 | 17.18 | 20.85 | 26.85 | 18.09 | 20.60 | 26.81 | 20.73 | 22.26 |
| NV | 28.33 | 22.58 | 26.08 | 27.78 | 24.22 | 23.93 | 30.71 | 24.79 | 26.05 |
| NeRF | 33.00 | 25.01 | 32.54 | 32.91 | 29.62 | 28.65 | 36.18 | 30.13 | 31.01 |
| NSVF | 33.19 | 25.18 | 32.29 | **34.27** | **32.68** | 27.93 | 37.14 | **31.23** | 31.74 |
| **GRF(Ours)** | **34.51** | **25.83** | **32.92** | 33.94 | 30.91 | **30.12** | **37.47** | 30.75 | **32.06** |
| | | | | SSIM↑ | | | | | |
| SRNs | 0.910 | 0.766 | 0.809 | 0.947 | 0.808 | 0.757 | 0.923 | 0.849 | 0.846 |
| NV | 0.916 | 0.873 | 0.880 | 0.946 | 0.888 | 0.784 | 0.944 | 0.910 | 0.893 |
| NeRF | 0.967 | 0.925 | 0.961 | 0.980 | 0.949 | 0.856 | 0.974 | 0.964 | 0.947 |
| NSVF | 0.968 | 0.931 | 0.960 | **0.987** | **0.973** | 0.854 | 0.980 | **0.973** | 0.953 |
| **GRF(Ours)** | **0.981** | **0.937** | **0.967** | **0.987** | 0.963 | **0.891** | **0.983** | 0.969 | **0.960** |
| | | | | LPIPS↓ | | | | | |
| SRNs | 0.106 | 0.267 | 0.200 | 0.063 | 0.174 | 0.299 | 0.100 | 0.149 | 0.170 |
| NV | 0.109 | 0.214 | 0.175 | 0.107 | 0.130 | 0.276 | 0.109 | 0.162 | 0.160 |
| NeRF | 0.046 | 0.091 | 0.050 | 0.028 | 0.063 | 0.206 | 0.121 | 0.044 | 0.081 |
| NSVF | 0.043 | 0.069 | **0.029** | **0.010** | **0.021** | 0.162 | **0.025** | **0.017** | **0.047** |
| **GRF(Ours)** | **0.021** | **0.068** | 0.042 | 0.013 | 0.041 | **0.141** | 0.028 | 0.032 | 0.048 |

Table 11: The PSNR, SSIM and LPIPS scores of our GRF simultaneously trained on 4 scenes of the Synthetic-NeRF dataset for multi-scene learning. The scores of SRNs, NeRF and NSVF trained on single scenes are included for comparison.

| | Chair | Mic | Ship | Hotdog |
|---|---|---|---|---|
| | | PSNR↑ | | |
| SRNs (Single-scene) | 26.96 | 26.85 | 20.60 | 26.81 |
| NeRF (Single-scene) | 33.00 | 32.91 | **28.65** | 36.18 |
| NSVF (Single-scene) | **33.19** | **34.27** | 27.93 | **37.14** |
| **GRF (Multi-scene)** | 32.49 | 32.02 | 27.76 | 34.92 |
| | | SSIM↑ | | |
| SRNs (Single-scene) | 0.910 | 0.947 | 0.757 | 0.923 |
| NeRF (Single-scene) | 0.967 | 0.980 | 0.856 | 0.974 |
| NSVF (Single-scene) | 0.968 | **0.987** | 0.854 | **0.980** |
| **GRF (Multi-scene)** | **0.971** | 0.982 | **0.866** | 0.975 |
| | | LPIPS↓ | | |
| SRNs (Single-scene) | 0.106 | 0.063 | 0.299 | 0.100 |
| NeRF (Single-scene) | 0.046 | 0.028 | 0.206 | 0.121 |
| NSVF (Single-scene) | 0.043 | **0.010** | **0.162** | **0.025** |
| **GRF (Multi-scene)** | **0.032** | 0.019 | 0.167 | 0.040 |

Table 12: The PSNR, SSIM and LPIPS scores of our GRF and NeRF on four novel scenes of Synthetic-NeRF in Group 1&2 experiments.

|  | Drums | Lego | Materials | Ficus | *mean* |
|---|---|---|---|---|---|
| PSNR↑ | | | | | |
| **GRF (Group 1)** | 13.23 | 13.53 | 12.26 | 15.47 | 13.62 |
| NeRF (Group 2, 100 iters) | 14.54 | 14.92 | 15.42 | 15.72 | 15.15 |
| NeRF (Group 2, 1k iters) | 18.01 | 20.04 | 20.40 | 20.81 | 19.81 |
| NeRF (Group 2, 10k iters) | 21.57 | 24.99 | 23.36 | 23.47 | 23.35 |
| **GRF (Group 2, 100 iters)** | 18.70 | 20.24 | 18.81 | 21.03 | 19.69 |
| **GRF (Group 2, 1k iters)** | 20.49 | 23.64 | 21.87 | 22.02 | 22.00 |
| **GRF (Group 2, 10k iters)** | 23.11 | 27.07 | 25.11 | 25.11 | 25.10 |
| SSIM↑ | | | | | |
| **GRF (Group 1)** | 0.762 | 0.736 | 0.703 | 0.849 | 0.763 |
| NeRF (Group 2, 100 iters) | 0.769 | 0.717 | 0.716 | 0.808 | 0.752 |
| NeRF (Group 2, 1k iters) | 0.793 | 0.775 | 0.812 | 0.857 | 0.809 |
| NeRF (Group 2, 10k iters) | 0.865 | 0.862 | 0.877 | 0.896 | 0.875 |
| **GRF (Group 2, 100 iters)** | 0.822 | 0.813 | 0.829 | 0.878 | 0.835 |
| **GRF (Group 2, 1k iters)** | 0.856 | 0.877 | 0.878 | 0.894 | 0.876 |
| **GRF (Group 2, 10k iters)** | 0.901 | 0.924 | 0.913 | 0.923 | 0.916 |
| LPIPS↓ | | | | | |
| **GRF (Group 1)** | 0.256 | 0.273 | 0.301 | 0.150 | 0.246 |
| NeRF (Group 2, 100 iters) | 0.332 | 0.395 | 0.314 | 0.393 | 0.359 |
| NeRF (Group 2, 1k iters) | 0.254 | 0.264 | 0.229 | 0.164 | 0.228 |
| NeRF (Group 2, 10k iters) | 0.157 | 0.154 | 0.128 | 0.110 | 0.137 |
| **GRF (Group 2, 100 iters)** | 0.196 | 0.203 | 0.159 | 0.117 | 0.169 |
| **GRF (Group 2, 1k iters)** | 0.154 | 0.138 | 0.123 | 0.097 | 0.128 |
| **GRF (Group 2, 10k iters)** | 0.104 | 0.090 | 0.090 | 0.071 | 0.089 |

## A.3   DETAILS OF EXPERIMENTAL RESULTS ON THE REAL-WORLD DATASET

Table 13: Comparison of the PSNR, SSIM and LPIPS scores of our GRF, SRNs (Sitzmann et al., 2019), LLFF (Mildenhall et al., 2019) and NeRF (Mildenhall et al., 2020) in the real-world dataset for single-scene learning.

|          | Room  | Fern  | Leaves | Fortress | Orchids | Flower | T-Rex | Horns | *Mean* |
|----------|-------|-------|--------|----------|---------|--------|-------|-------|--------|
|          |       |       |        | PSNR↑    |         |        |       |       |        |
| SRNs     | 27.29 | 21.37 | 18.24  | 26.63    | 17.37   | 24.63  | 22.87 | 24.33 | 22.84  |
| LLFF     | 28.42 | 22.85 | 19.52  | 29.40    | 18.52   | 25.46  | 24.15 | 24.70 | 24.13  |
| NeRF     | **32.70** | 25.17 | 20.92 | 31.16 | 20.36 | 27.40 | 26.80 | 27.45 | 26.50 |
| **GRF(Ours)** | 31.74 | **25.72** | **21.16** | **31.28** | **20.88** | **27.83** | **27.01** | **27.50** | **26.64** |
|          |       |       |        | SSIM↑    |         |        |       |       |        |
| SRNs     | 0.883 | 0.611 | 0.520  | 0.641    | 0.449   | 0.738  | 0.761 | 0.742 | 0.668  |
| LLFF     | 0.932 | 0.753 | 0.697  | 0.872    | 0.588   | 0.844  | 0.857 | 0.840 | 0.798  |
| NeRF     | 0.948 | 0.792 | 0.690  | 0.881    | 0.641   | 0.827  | 0.880 | 0.828 | 0.811  |
| **GRF(Ours)** | **0.951** | **0.827** | **0.727** | **0.898** | **0.667** | **0.852** | **0.901** | **0.873** | **0.837** |
|          |       |       |        | LPIPS↓   |         |        |       |       |        |
| SRNs     | 0.240 | 0.459 | 0.440  | 0.453    | 0.467   | 0.288  | 0.298 | 0.376 | 0.378  |
| LLFF     | 0.155 | 0.247 | **0.216** | 0.173 | 0.313 | **0.174** | 0.222 | 0.193 | 0.212 |
| NeRF     | 0.178 | 0.280 | 0.316  | 0.171    | 0.321   | 0.219  | 0.249 | 0.268 | 0.250  |
| **GRF(Ours)** | **0.104** | **0.191** | 0.238 | **0.127** | **0.275** | 0.176 | **0.146** | **0.169** | **0.178** |

### A.4 ANALYSIS OF ATTENTION MECHANISM

The attention mechanism in our GRF aims to automatically select the correct pixel patch from multiple pixel patches where the light rays intersect at the same query 3D point in space.

In order to investigate how the attention mechanism learns to select the useful information, we retrieve the maximal attention score from the observed multiple pixel patches for analysis. Intuitively, the higher the attention score is assigned to a particular pixel patch, the more important that patch for inferring the novel pixel RGB. In particular, we conduct the following experiment using our GRF model trained on ShapeNetv2 Cars. In this case, the AttSets attention module is used (details are in Table 9). Given a query light ray, multiple 3D points are sampled to query the network.

- We firstly try to find the 3D point which is near the surface according to the predicted volume density for points along the ray through a given pixel, if they exist. Otherwise, we ignore such pixels, making them white.
- Then we compute the M feature vectors from the input M views for these surface points.
- Thirdly, the attention masks for those M feature vectors are computed. We identify the view whose sum of the attention mask along the feature axis is greatest as the main contributor for inferring the novel pixel RGB.
- After querying light rays for each pixel, we obtain a rendered RGB image. At the same time, for each pixel of that image, we select the most important view from the M input views for the surface-intersection point along the ray from the viewpoint through that pixel, according to the maximal attention score. Eventually, we obtain a Max Attention Map corresponding to the rendered RGB image.

Figure 7 shows the qualitative results of the above experiment. In particular, we feed the three images (#1,#2,#3) of an unseen car into our GRF model which is well-trained on car category, and then render a new image (e.g., the 5th image in Figure 7). Note that, we carefully select the input 3 images and the rendered image with very large viewing baselines. In the mean time, we obtain and visualize the Max Attention Map corresponding to the rendered image.

For each pixel of the rendered image, we retrieve the input image pixel that has the highest attention score. Specifically, the rendered pixels with purple color correspond to the input image #1, the green pixels correspond to the input image #2, while the blue pixels correspond to the input image #3.

**Analysis.** It can be seen that, when inferring a new image, the attention module of our GRF focuses on the most informative pixel patch from the multiple input pixel patches. In addition, it is able to truly deal with the visual occlusion. For example, when inferring the windshield of the car, the attention module focuses on the input image #2 where the windshield is visible, while ignoring the image #1 and #3 where the windshield is self-occluded.

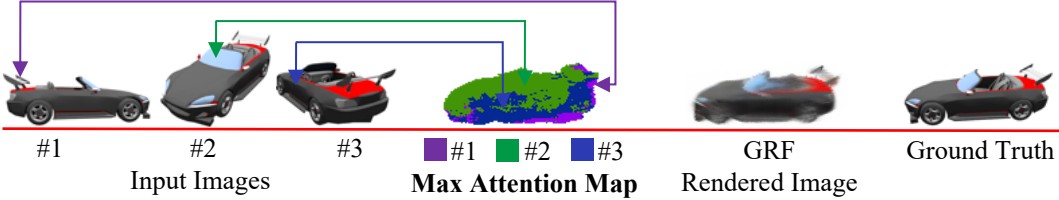

Figure 7: Visualization of Max Attention Map from Multiple Input Images of a Novel Object for Inferring a Novel View.

## A.5 GENERALIZATION TO VISUAL OCCLUSIONS AND VARIABLE INPUT IMAGES

We carefully select the attention module, i.e., either AttSets or SlotAtt, to aggregate the features from an arbitrary number of input views. In order to evaluate how our GRF is able to generalize with a variable number of input views, especially when there is a very sparse number of views with severe visual occlusions, we conduct the following four groups of experiments.

- **1-view Reconstruction**. We feed the a single image of a novel car into our GRF model which is well-trained on car category (trained with 5 images per object), and then render 9 new images from vastly different viewing angles. This is the extreme case where the majority of the object is self-occluded.
- **2-view Reconstruction**. Similarly, we feed only two images of the novel car into the same model and render the same 9 novel views. In this case, more information is given to the network, but there are still many parts occluded.
- **5-view / 10-view Reconstruction**. The same GRF model is fed with 5 and 10 views of the novel object, rendering the same set of new images.

**Analysis.** Figure 8 shows the qualitative results. It can be seen that: 1) In the extreme case, i.e., 1-view reconstruction, our GRF is still able to recover the general 3D shape of the unseen object, including the visually occluded parts, primarily because our CNN model learns the hierarchical features including the high-level shapes. 2) Given more input views, the originally occluded parts tend to be observed from some viewing angles, and then these parts can be reconstructed better and better. This shows that our GRF is indeed able to effectively identify the corresponding useful pixel features for more accurately recovering shape and appearance.

Figure 8: Qualitative results of our GRF when being fed with a variable number of views of a novel object. The red circle highlights that the tail of the car is able to be recovered given more visual cues from more input images.

## A.6 MORE QUALITATIVE RESULTS

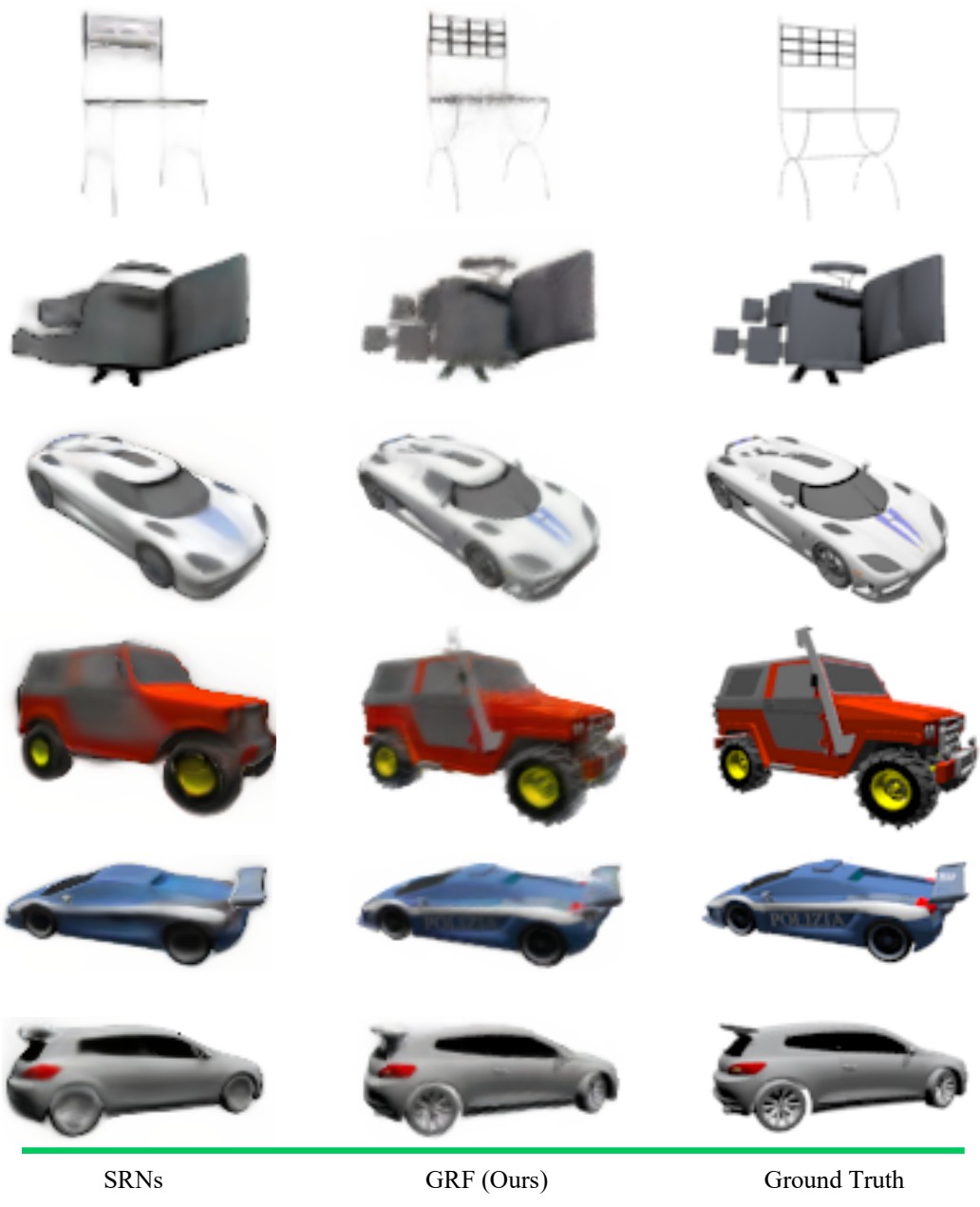

| SRNs | GRF (Ours) | Ground Truth |

Figure 9: Qualitative results of SRNs and our GRF for 50-view reconstruction of chairs and cars.

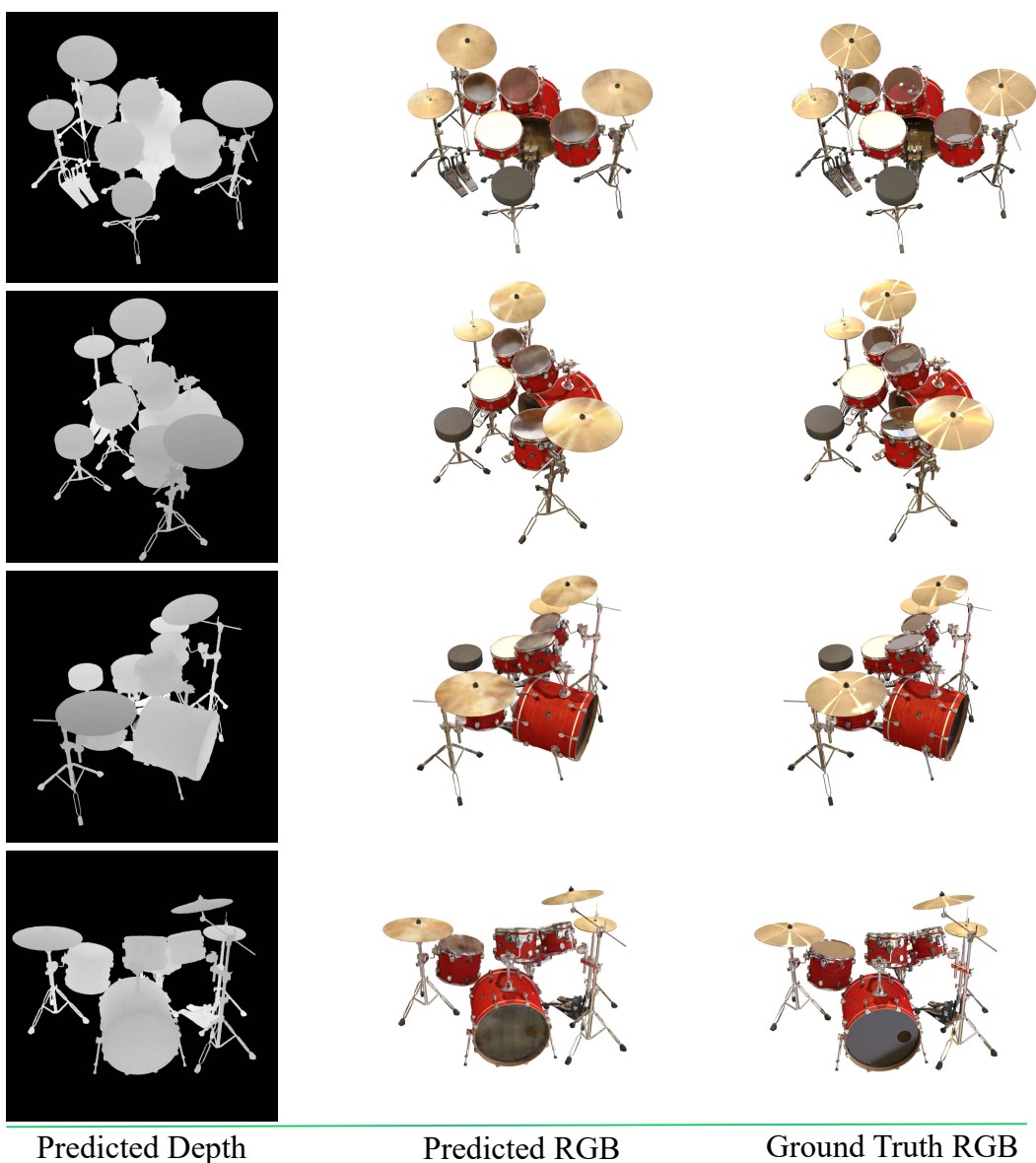

Predicted Depth       Predicted RGB       Ground Truth RGB

Figure 10: Qualitative results of our GRF for novel view depth and RGB estimation on the Synthetic-NeRF dataset.

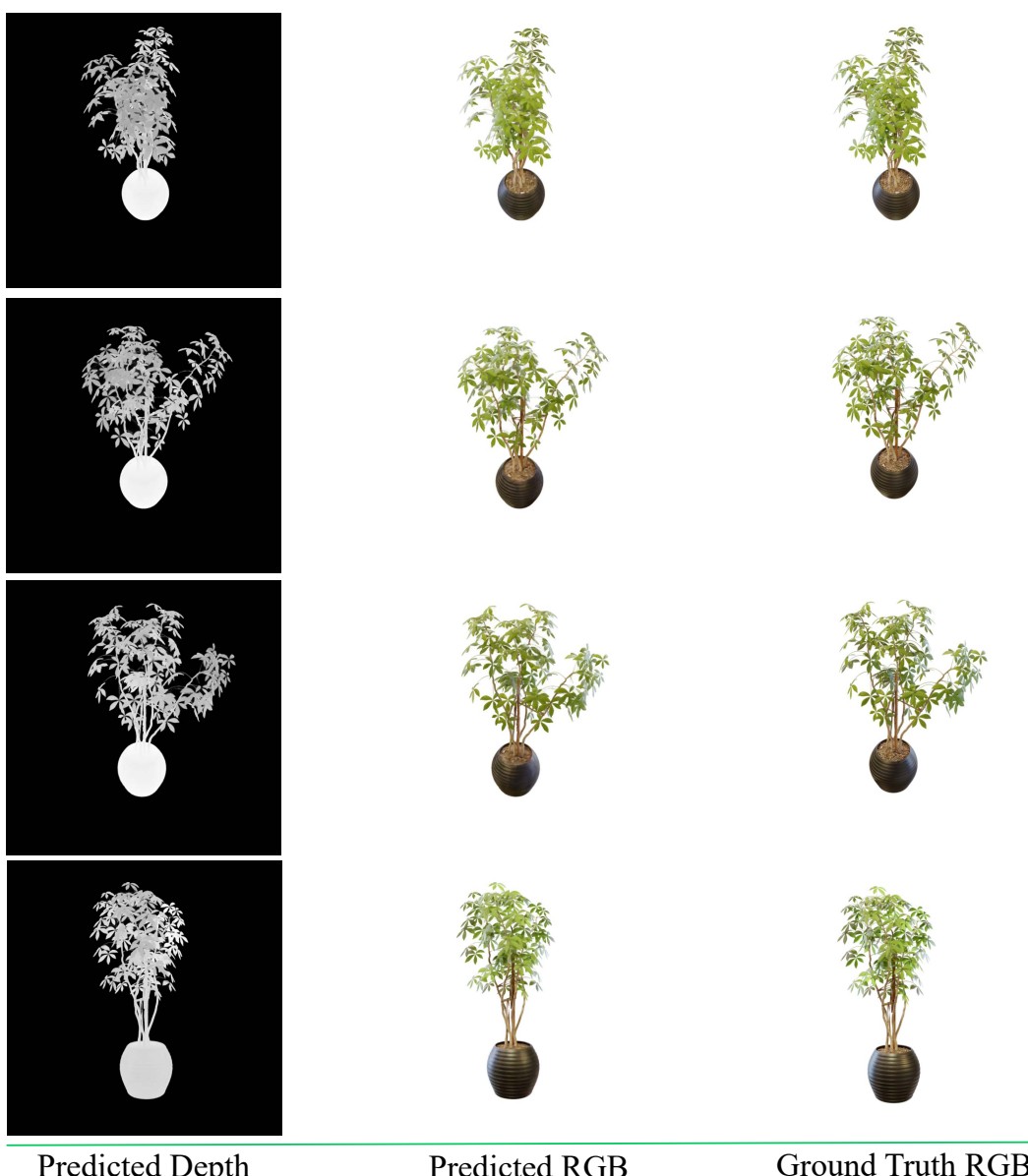

| Predicted Depth | Predicted RGB | Ground Truth RGB |

Figure 11: Qualitative results of our GRF for novel view depth and RGB estimation on the Synthetic-NeRF dataset.

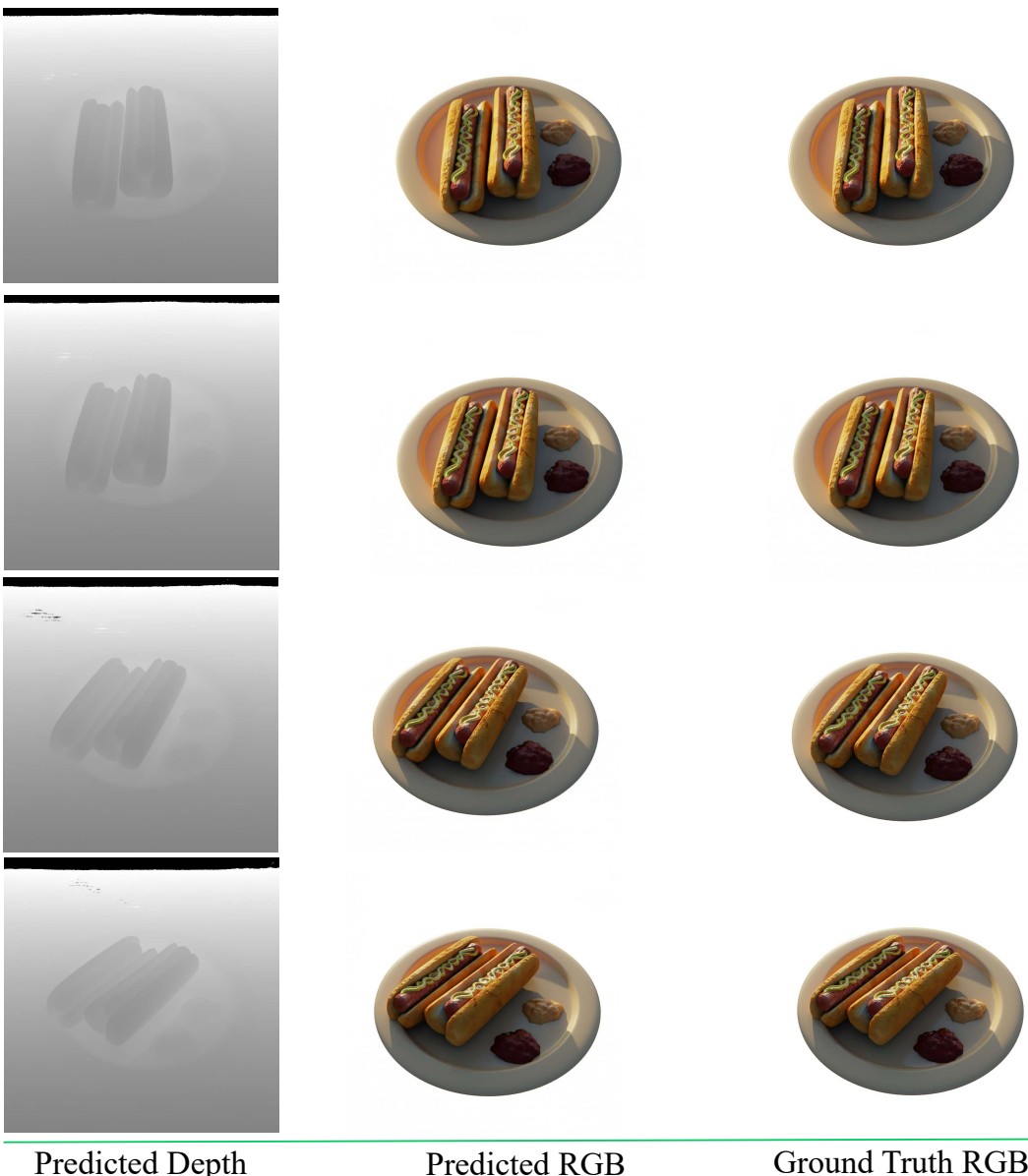

Predicted Depth          Predicted RGB          Ground Truth RGB

Figure 12: Qualitative results of our GRF for novel view depth and RGB estimation on the Synthetic-NeRF dataset.

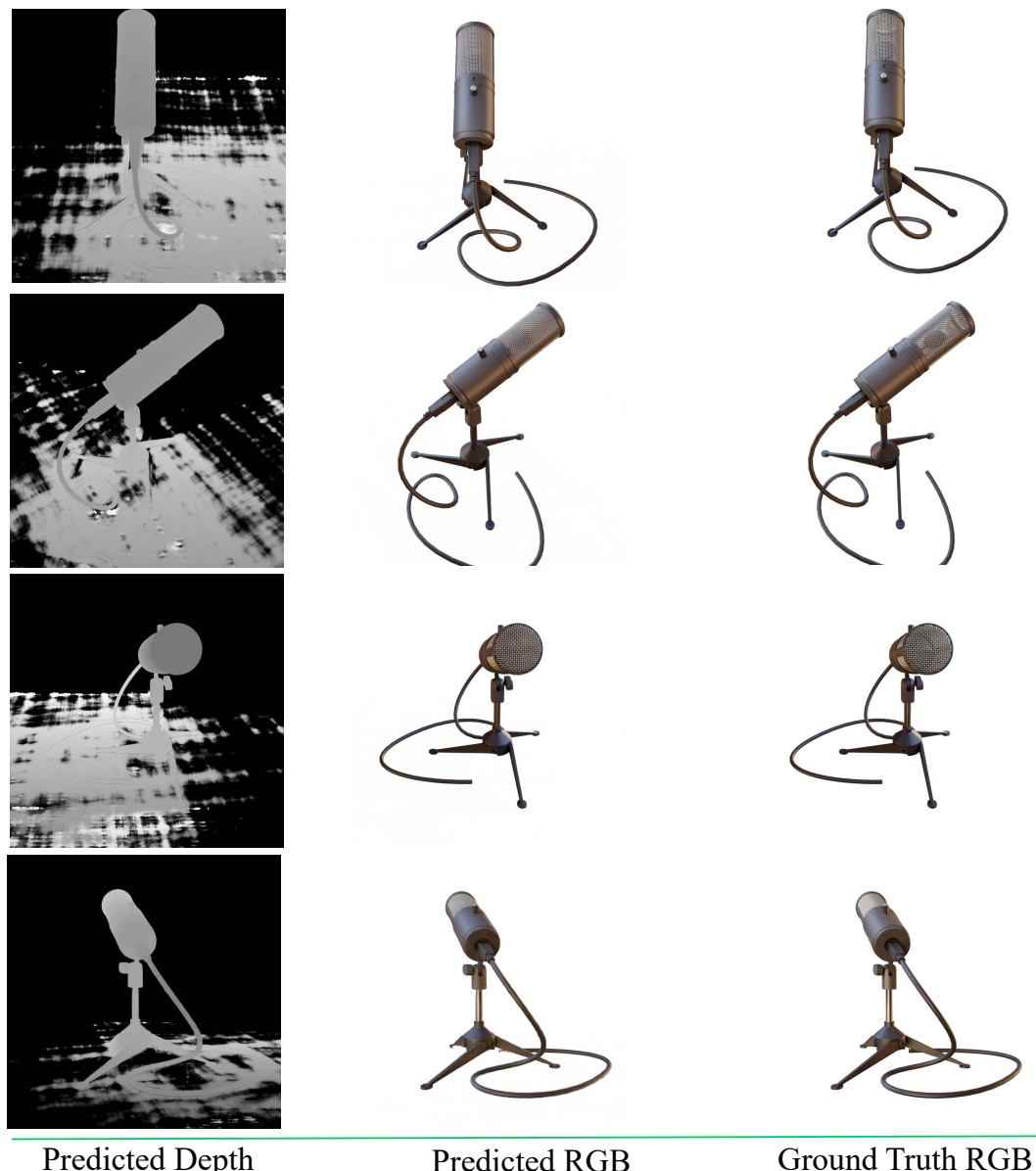

Predicted Depth        Predicted RGB        Ground Truth RGB

Figure 13: Qualitative results of our GRF for novel view depth and RGB estimation on the Synthetic-NeRF dataset.

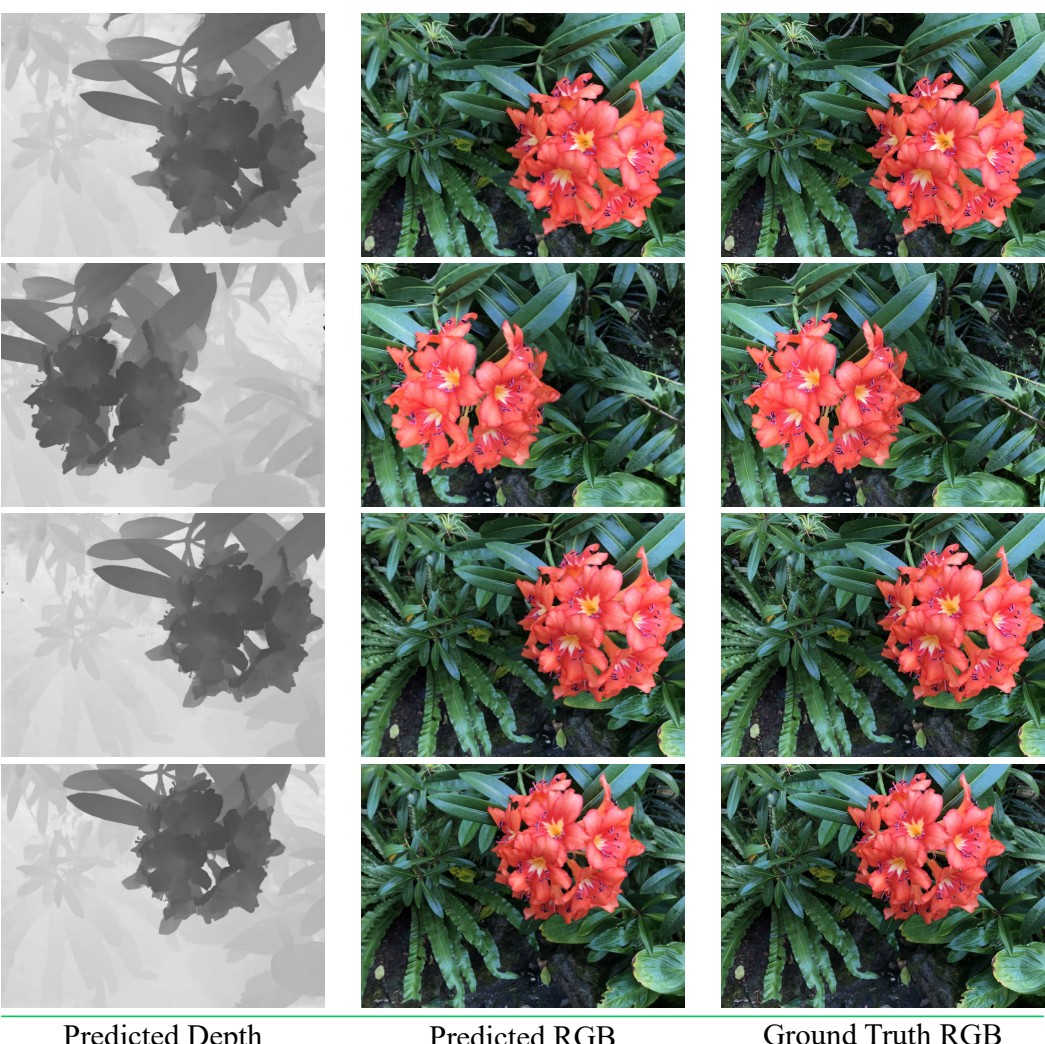

Predicted Depth          Predicted RGB          Ground Truth RGB

Figure 14: Qualitative results of our GRF for novel view depth and RGB estimation on the real-world dataset.

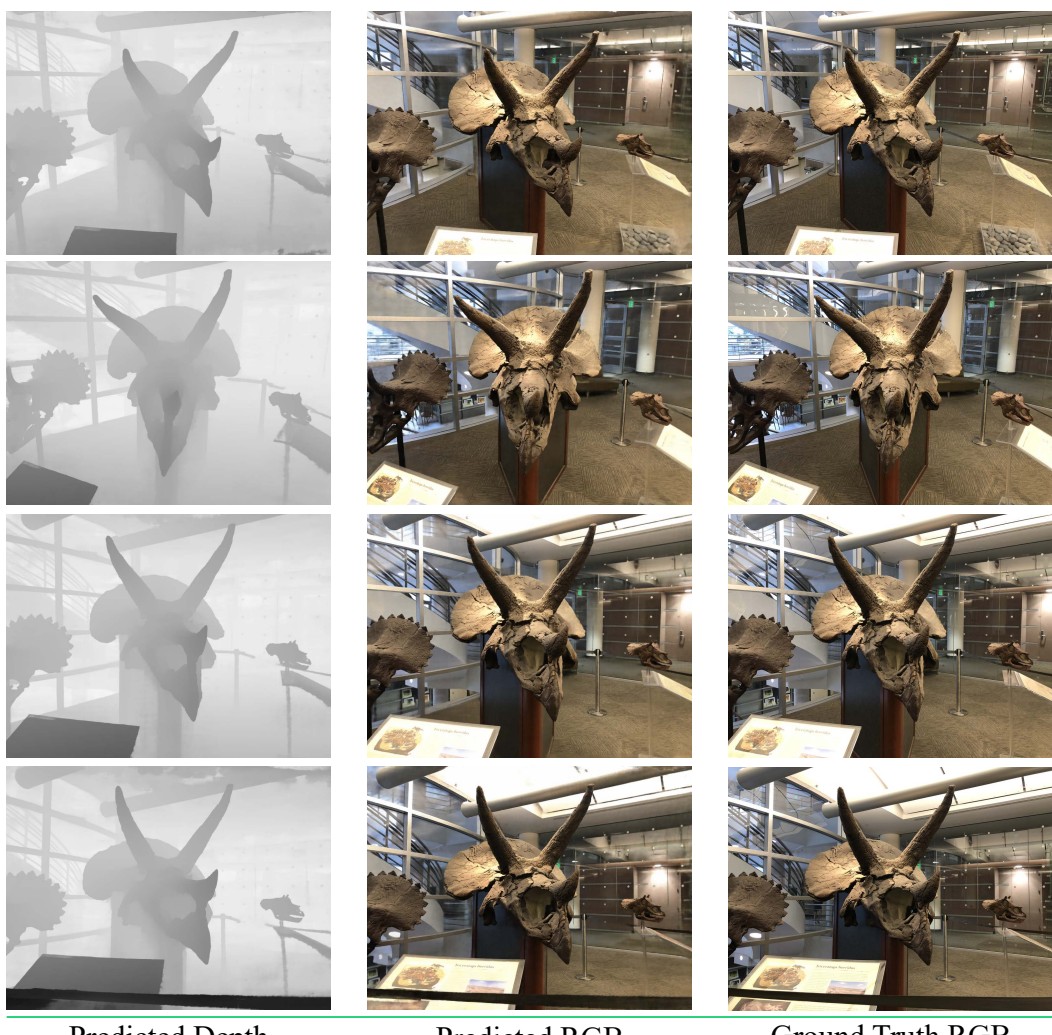

Predicted Depth             Predicted RGB             Ground Truth RGB

Figure 15: Qualitative results of our GRF for novel view depth and RGB estimation on the real-world dataset.

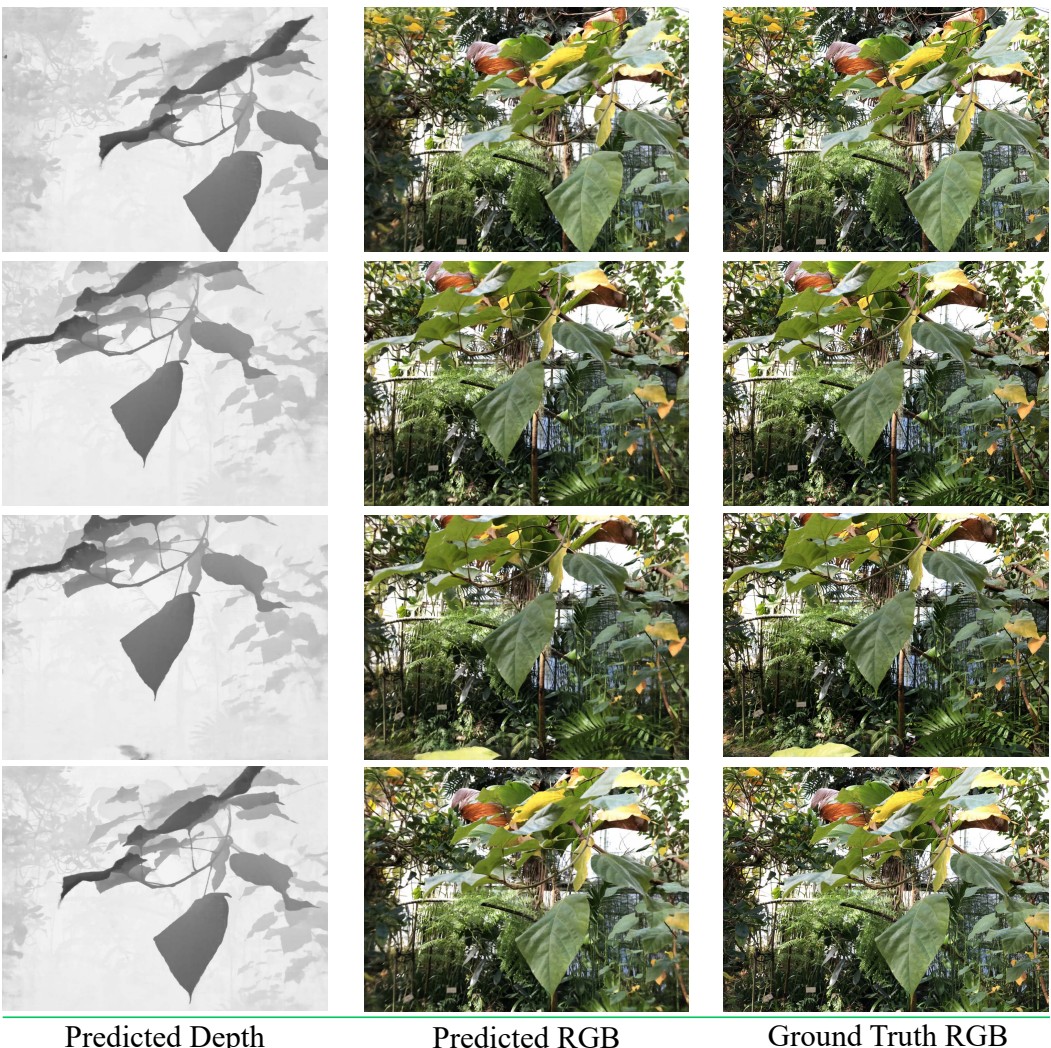

Predicted Depth        Predicted RGB        Ground Truth RGB

Figure 16: Qualitative results of our GRF for novel view depth and RGB estimation on the real-world dataset.

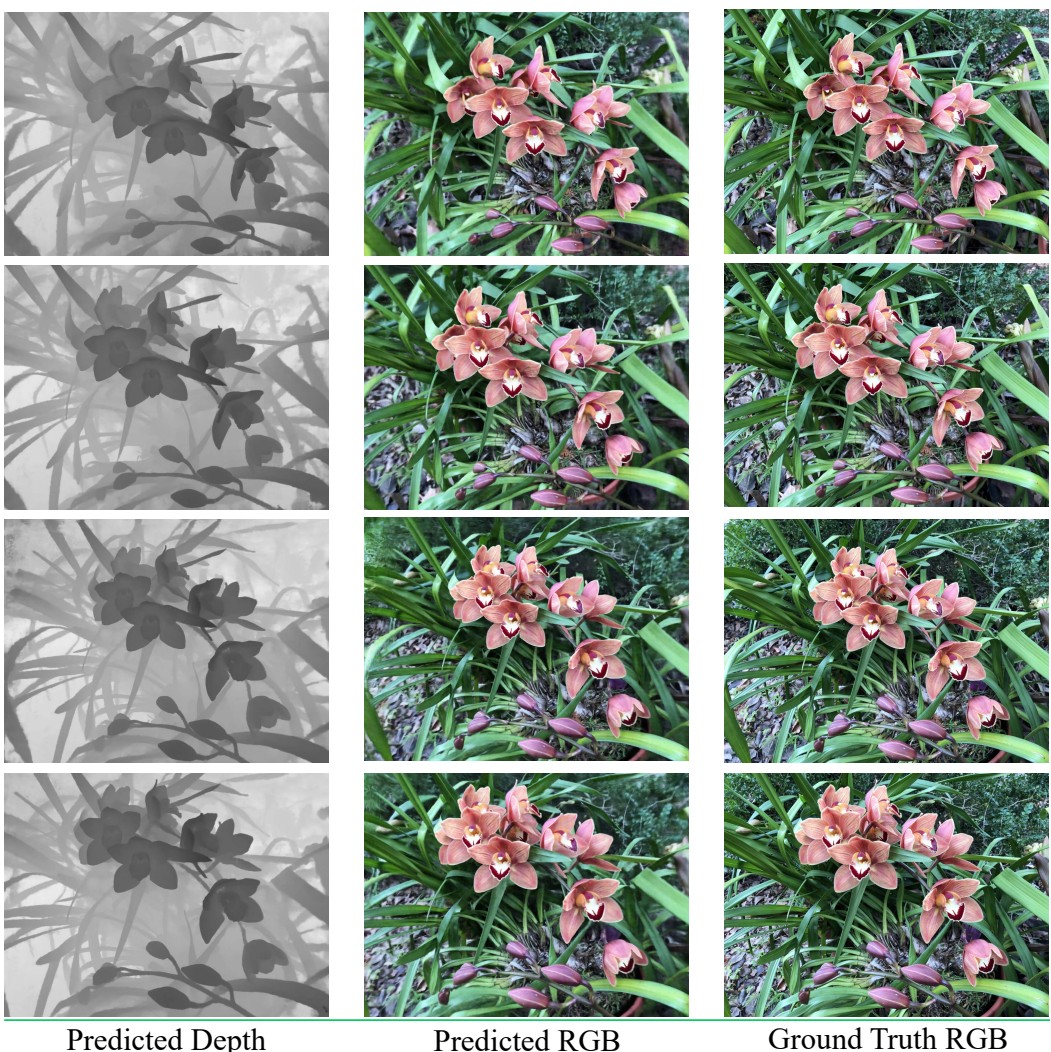

Predicted Depth          Predicted RGB          Ground Truth RGB

Figure 17: Qualitative results of our GRF for novel view depth and RGB estimation on the real-world dataset.

