# OpenReview forum: "GRF: Learning a General Radiance Field for 3D Scene Representation and Rendering"
_ICLR.cc/2021/Conference — Reject_

### Official Review · AnonReviewer2 · 2020-10-26
**Recommendation to weakly Reject**

**Rating:** 4
**Confidence:** 3

**Review:**

Summary:

The paper proposed an extension on Neural radiance field for better generalization across novel scenes by introducing multi-view pixel aligned features as additional input to NeRF. To fuse multi-view information and implicit reason about occlusion, an attention aggregation module is applied.

Strengths:

+ The idea of using pixel aligned feature for making NeRF generalize to novel scenes is interesting.

Concerns:

1) the term of implicit representation is interchangeably used to describe implicit function and explicit 3D representation parameterized by MLPs.

2) Comparison between GRF and GRAF: In GRAF, the radiance field is also conditioned on a shape code as well as an appearance code. But in the related work section, the author states that GRAF is unable to generalize to novel scenarios which seems to be an unfair claim.

3) Missing details about volumetric rendering: The paper did not talk about how rendering is performed(possibly volumetric rendering). And both in NeRF and NSVF, sampling strategy and volumetric rendering both play important roles on achieving high-fidelity rendering results. It is unclear here how ray marching is formulated.

4) Experiments on ShapeNetV2: the author just reported SRN's result here, but didn't compare with some other method like NVS which is also directly applicable. Besides, other methods like NeRF could be modified to train on multiple objects like including a conditional embedding to NeRF jsut like GRAF or like in SRN use hyper-networks. Lack results here could be potentially undermining the claim of GRF being more general and robust.

5) Experiments on real-world complex scenes: The training setup here is a bit unclear to me. From just the description there, it is hard for me to tell whether GRF is training one model on all those scenes or training separate models for different scenes. It would be great if the author could make that clear.

Minors:

In the last paragraph of section 3.4, very query 3D point -> every query 3D  point?

---

> ### Author Response · Authors · 2020-11-24
> **Response to Reviewer 2**
>
> **Concern 1**\
> The term of implicit representation is interchangeably used to
> describe implicit function and explicit 3D representation parameterized
> by MLPs.\
> **Response**: \
> We use implicit function to describe the network architecture itself,
> while implicit representation to describe the learned features in
> general. More information about these terms may be found in the paper
> "Local Deep Implicit Functions for 3D Shape, CVPR'20". In the revised
> paper, we have carefully checked the consistency of the terms.
> ------------------------------------------------------------------------
> **Concern 2**\
> Comparison between GRF and GRAF: In GRAF, the radiance field is also
> conditioned on a shape code as well as an appearance code. But in the
> related work section, the author states that GRAF is unable to
> generalize to novel scenarios which seems to be an unfair claim.\
> **Response**:\
> Thanks for the advice. We have rephrased the sentences accordingly.
> ------------------------------------------------------------------------
> **Concern 3**\
> Missing details about volumetric rendering: The paper did not talk
> about how rendering is performed(possibly volumetric rendering). And
> both in NeRF and NSVF, sampling strategy and volumetric rendering both
> play important roles on achieving high-fidelity rendering results. It is
> unclear here how ray marching is formulated.\
> **Response**:\
> In the revised paper, we have updated the section 3.5, clearly
> specifying that we strictly follow NeRF for the sampling strategy and
> volumetric rendering.
> ------------------------------------------------------------------------
> **Concern 4**\
>  Experiments on ShapeNetV2: the author just reported SRN's result
> here, but didn't compare with some other method like NVS which is also
> directly applicable. Besides, other methods like NeRF could be modified
> to train on multiple objects like including a conditional embedding to
> NeRF jsut like GRAF or like in SRN use hyper-networks. Lack results here
> could be potentially undermining the claim of GRF being more general and
> robust.\
> **Response**:\
> These are helpful points. On the ShapeNetV2 (cars and chairs) dataset,
> SRNs is currently the state-of-the-art approach for novel view
> generation. To the best of our knowledge, there is no other NVS (novel
> view synthesis) method reporting results on this particular dataset.
> Although some methods may be trained from scratch on this dataset, it is
> non-trivial to conduct the experiments considering the extremely
> expensive computation.
>
> It is insightful to point out that the conditional embedding to NeRF can
> be used for comparison. In fact, in our ablation study, we analyze that
> the removal of Point Local Features is a baseline of \"Conditional
> embedding + NeRF\". In the revised paper, we update the section 4.4
> accordingly.
> ------------------------------------------------------------------------
> **Concern 5**\
>  Experiments on real-world complex scenes: The training setup here is
> a bit unclear to me. From just the description there, it is hard for me
> to tell whether GRF is training one model on all those scenes or
> training separate models for different scenes. It would be great if the
> author could make that clear.\
> **Response**:\
> We thank the reviewer for pointing out this ambiguity. We have changed
> the language to make it explicit where single-scene learning is
> happening, and where multi-scene learning is happening.

---

> > ### Comment · AnonReviewer2 · 2020-11-25
> > **Remain my score of rejection**
> >
> > Dear author(s),
> >
> >  Thanks you for your detailed replies. However, I still think some of my concerns are not well addressed. The main reason is still lack or more experimental results to support the argument made in this manuscript. In general, I think the idea of using pixel aligned feature for better scene representations is very interesting. Nevertheless, the experiment part and the writing about methodology still needs to be improved. For example, it is still quite confusing for me to understand the result and setting of group 2 on novel-scene generalization. How is NeRF trained for those novel scenes? Is a single nerf trained to fit all the novel scenes or several nerf are trained to fit separate scenes? If multiple models are trained, the results seem to be a little bit lower comparing to the results reported in the original NeRF paper. Also, it is not very convincing to show that GRF is better in generalization than NeRF when NeRF is not converging. On the other hand, it would be great to include one baseline like what did in the GRAF paper, like using some other kind of conditional input or like SRN using a hyper-network to predict the parameter for different NeRF network.
> >
> > One other comment is still about the implicit function term. As stated in NeRF, their paper are using zero term of implicit function and they make it every clear that they have learnt a continuous volumetric function which is not implicit at all comparing to a sign-distance field or other similar representations. Please also refer to https://en.wikipedia.org/wiki/Implicit_function for a more detailed definition of implicit function.

---

> > > ### Author Response · Authors · 2020-11-25
> > > **We highly appreciate but strongly disagree with the comments**
> > >
> > > **Concern 1**\
> > > Thanks you for your detailed replies. However, I still think some of my concerns are not well addressed. The main reason is still lack or more experimental results to support the argument made in this ... ... and the writing about methodology still needs to be improved.
> > >
> > > For example, it is still quite confusing for me to understand the result and setting of group 2 on novel-scene generalization. How is NeRF trained for those novel scenes? Is a single nerf trained to fit all the novel scenes or several nerf are trained to fit separate scenes?
> > >
> > > **Response**\
> > > Thank you for the comments. In our revised paper, extensive experimental results (quantitative and qualitative) are provided to clearly support, 1) the novelty and contribution of our GRF, 2) the advantage of our GRF over SRNs, 3) the advantage of our GRF over original NeRF, and 4) the effectiveness of the CNN module and attention module. All arguments in the entire paper are well presented and validated.
> > >
> > > As to the experiment Group 2, to quote our sentences directly (sec 4.2, page 7 and 8), "For comparison, we also train NeRF on each of the four novel scenes with 100, 1k, 10k iterations from scratch." Note that, NeRF can only be trained on a single scene at a time.
> > >
> > > **Concern 2**\
> > > If multiple models are trained, the results seem to be a little bit lower comparing to the results reported in the original NeRF paper. Also, it is not very convincing to show that GRF is better in generalization than NeRF when NeRF is not converging.
> > >
> > > **Response**\
> > > Fundamentally, as we clearly specify in the paper, the experiments in Group 2 are to validate that our GRF can learn faithful geometries extremely quickly given the same number of supervision signals, thanks to the generality of the features learned from the other 4 scenes. To the best of our knowledge, there is no method that can generate faithful novel views of out-of-distribution objects from high-resolution images (in a single forward pass or just a few iterations of training) as our GRF can here.
> > >
> > > The reviewer may miss this key point of our experiments, thereby saying that "... not convincing when NeRF is not converging". This is irrelevant to our experiments.
> > >
> > >
> > > **Concern 3**\
> > > On the other hand, it would be great to include one baseline like what did in the GRAF paper, like using some other kind of conditional input or like SRN using a hyper-network to predict the parameter for different NeRF network.
> > >
> > > **Response**\
> > > As we clearly responded earlier and discussed in the revised paper, the section 4.4, page 9, our ablated model of "Remove Pixel Local Features" is a "conditional embedding + NeRF network", where the global features for each object in cars are learned. This ablation performs SEVEN psnr worse, fundamentally because such feature embeddings can never encode the rich and precise pixel-level local features learned by GRF for such a huge number of different scenes.
> > >
> > > To sum up, we thank the reviewer again for this repeated comment, but it has already been fulfilled in our manuscript.
> > >
> > >
> > > **Concern 4**\
> > > One other comment is still about the implicit function term. ... .... function and they make it every clear that they have learnt a continuous volumetric function which is not implicit at all comparing to a sign-distance field or ........ definition of implicit function.
> > >
> > > **Response**\
> > > There are always technique terms that are widely used in a specific area, while they may not have exactly the same meaning from the perspective of different researchers.
> > >
> > > As the reviewer points out, NeRF learns a continuous volumetric function, while DeepSDF (CVPR'19) learns a continuous signed distance function. The inputs of both methods are the location of 3D query point (i.e, xyz), except that they have different meanings for the outputs. Basically, both can be called implicit functions, primarily because the word IMPLICIT here is widely used to differentiate from the EXPLICIT 3D functions such as voxel grids, point clouds, and meshes. This may be a bit confusing for the audience out of the field of 3D vision and learning. The cited wiki page is interesting, but the pure mathematical definition looks a bit far from the domain of 3D scene representation.
> > >
> > > However, to avoid possible confusion, we are happy to replace the term "implicit function" with "neural function for implicit representation", if it is suggested to do so.
> > >
> > > **Final**
> > >
> > > Overall, we truly appreciate all the efforts and feedback from the reviewer, and we enjoy a lot in the discussion. However, it seems that the expertise of the reviewer may not exactly fall in the area of our manuscript, according to the interesting and surprising comments. Also, the lower confidence score reported by the reviewer seems consistent with this interpretation.
> > >
> > > We hope our responses are helpful for the audience, peer reviewers, and area chairs to have further discussion about the contribution of our manuscript to the community.

---

### Official Review · AnonReviewer1 · 2020-10-27
**A generic scene radiance field neural estimator with better generalization ability than NeRF.   Well motivated idea but poorly executed.**

**Rating:** 5
**Confidence:** 4

**Review:**

This paper presents a new neural learner for learning generic scene radiance function. Compared with existing methods such as NeRf which are scene-specific requiring per scene based training, the proposed new method is potentially able to generalize to novel unseen scenes or objects.   Specifically, the method represents scene as a neural radiance filed in terms of spatial coordinates of camera centre and a query point. An attention model is used for aggregating back-projected image features at every query location, and the aggregated feature vectors are then used for predicting its RGB-alpha radiance.

The work is well motivated, aiming to relax the restriction of existing per-scene based radiance field learning methods  (NeRF, for example).  Experiments show some improved performance in novel view/radiance field synthesis , however I do no f find there are convincing tests provided or conducted to validate the "better generalizability" claim.   Overall, the current results are insufficient to validate the method's generality to novel scenes .More comments are given below:

Pros:
+ The work is well motivated.
+ The idea of using attention model to address multi-view consistency issue is interesting,  which appears to be sound and promising.
+ extensive experimental study, lots of ablation studies and comparisons.

Cons:
-One key intention of this paper is to improve the network's generalibility to unseen, novel scenes.   However, throughout the reported experiments,   I do not find  adequate experimental evidences to  support this claim.    From the paper it seems the proposed method has only been tested on some unseen scenes in the ShapeNet dataset, however with very similarly-looking objects in simple and similar poses.    What are the results on the other two datasets, and what are the training-testing split on unfamiliar scenes?  Since the major motivation of this paper is to handle complex novel unseen scenes, I would suggest the authors conduct experiments on outdoor scenarios, such as the 'Tanks' and 'Temples' dataset commonly used in recent related work on neural novel view scene rendering.

- It is mentioned that two different attention models (AttSets and Slot Attention) are used depending on training dataset. However it is unclear which model was used for testing which datasets.  How did you compare in PSNR/SSIM metrics on each dataset?   Also, there is no description of how these attention mechanisms help the process.  In particular, since 3D point visibility of source view features under target view is one of the key problems for any novel view synthesis work.  How the employed attention module solves this issue is unclear.  In short,  I did not see how this attention module handles the visibility problem, other than (as the authors said) it helps to " aggregate multiple source view features"-- which seems to me to be an obvious outcome.
- The authors  also claimed that the attention module is invariant to the input view number and orders;  Yet,  there is not any  evidence (either theoretic analysis,  or experimental) provided in the paper.   Is it an inherent property of the Attention modules ?

- From reading Table-1,  it seems SRNs sometimes outperforms GRF.   The advantage of the proposed GRF is not entirely clear.

- overall, a well motivated paper, but the execution of the ideas is not convincing.

---

> ### Author Response · Authors · 2020-11-24
> **Response to Reviewer 1**
>
> **Concern 1**\
> One key intention of this paper is to improve the network's
> generalibility to unseen, novel scenes. However, throughout the reported
> experiments, I do not find adequate experimental evidences to support
> this claim. From the paper it seems the proposed method has only been
> tested on some unseen scenes in the ShapeNet dataset, however with very
> similarly-looking objects in simple and similar poses.
>
> What are the results on the other two datasets, and what are the
> training-testing split on unfamiliar scenes?
>
> Since the major motivation of this paper is to handle complex novel
> unseen scenes, I would suggest the authors conduct experiments on
> outdoor scenarios, such as the 'Tanks' and 'Temples' dataset commonly
> used in recent related work on neural novel view scene rendering.
>
> **Response**:\
> These points are highly valuable for our paper. As requested, we
> conducted additional experiments on the Synthetic-NeRF dataset to
> thoroughly evaluate the generalization capability of our GRF across
> novel scenes.
>
> In particular, we train a single GRF model on four different scenes, and
> then directly test it on the remaining completely different four scenes.
> The results show that our GRF can indeed generalize well across novel
> scenes.
>
> In addition, we conduct extra experiments to demonstrate that the
> learned GRF can greatly benefit the single-scene learning, significantly
> better than the NeRF trained from scratch given a sparse amount of
> training signals. All the details are presented in Section 4.2, page 7
> and 8.
>
> As to the suggested large-scale 3D scenes (Tanks and Temples), it would
> be interesting to explore. However, it is non-trivial to directly
> evaluate our GRF on it, because of the complexity of geometry and the
> expensive computation required. In the revised paper, this is fully
> discussed in section 4.3, the last paragraph of page 8, and we leave it
> for future exploration.
>
> ------------------------------------------------------------------------
> **Concern 2**\
> It is mentioned that two different attention models (AttSets and Slot
> Attention) are used depending on training dataset. However it is unclear
> which model was used for testing which datasets. How did you compare in
> PSNR/SSIM metrics on each dataset?\
> **Response**:\
> As noted in the appendix, we use AttSets for the ShapeNet dataset, and
> Slot Attention for the synthetic/real-world datasets. We use AttSets in
> the former because it can quickly aggregate information from multiple
> views, allowing the rendering of the millions of images needed for the
> ShapeNet benchmark. We also empirically find that the models for
> synthetic/real-world datasets converge quicker with Slot Attention.
> Comparison of the PSNR/SSIM metrics all strictly follows the baselines
> such as NeRF.
> ------------------------------------------------------------------------
> **Concern 3**\
> Also, there is no description of how these attention mechanisms help the
> process. In particular, since 3D point visibility of source view
> features under target view is one of the key problems for any novel view
> synthesis work. How the employed attention module solves this issue is
> unclear. In short, I did not see how this attention module handles the
> visibility problem, other than (as the authors said) it helps to \"
> aggregate multiple source view features\"-- which seems to me to be an
> obvious outcome.
>
> **Response**:\
> These are insightful questions. To thoroughly investigate how the
> attention module aids the network to deal with the visual occlusion, we
> conduct additional experiments and analyze in depth in the appendix
> (A.4). In particular, we track the largest attention score of the pixel
> patch and visualize the maximal attention score map for every pixel of a
> rendered image. Our results show that the attention module can indeed
> deal with the visual occlusion as it is expected.
> ------------------------------------------------------------------------
> **Concern 4**\
> The authors also claimed that the attention module is invariant to the
> input view number and orders; Yet, there is not any evidence (either
> theoretic analysis, or experimental) provided in the paper. Is it an
> inherent property of the Attention modules ?
>
> **Response**:\
> The property of being invariant to the input views is theoretically
> analyzed in the papers AttSets and Slot-Att. To avoid the confusion, we
> rephrased the sentence in the revised paper.
> ------------------------------------------------------------------------
> **Concern 5**\
> From reading Table-1, it seems SRNs sometimes outperforms GRF. The
> advantage of the proposed GRF is not entirely clear.
>
> **Response**:\
> In the revised paper, we have added additional results in section 4.1,
> page 6 and 7. We show that SRNs cannot generalize at all without
> retraining on novel scenes, while our GRF can.

---

> ### Author Response · Authors · 2020-11-25
> **Response to Reviewer's Comments**
>
> Dear R#1:
>
> We thank you for careful reading through all our responses and the revised paper. It is also great to receive your comments back again. We would like to take this opportunity to further discuss your remaining concerns and clarify the novelty/contribution of our paper.
>
> **1**\
> Remaining Concerns. We agree that your initial requirements are not all fulfilled, especially the possible experiments on the large-scale 3D scene dataset "Tanks and Temples". Yet, we fully discussed the reasons in the paper why our GRF is somehow limited to achieve such super exciting technical maturity. Moreover, we have provided compelling evidence for the generality of our approach in the revised paper. Apart from that, we would also like to kindly ask what are the remaining concerns regarding possible inaccurate arguments in the paper. We are definitely happy to further improve it accordingly, and we believe this is achievable considering that there is enough time to incorporate them into the final version.
>
> **2**\
> Our Novelty/Contribution. To the best of our knowledge, following the great success of NeRF, there are two papers successfully published very recently, NSVF (NeurIPS'20 spotlight) and GRAF (NeurIPS'20). NSVF uses sparse voxels, along with the original continuous 3D points, to represent 3D scenes, significantly improving the querying speed and efficiency. GRAF leverages the powerful rendering capability of NeRF to build a great
>  GENERATIVE model which is able to maintain the geometry. However, neither NVSF nor GRAF is extending NeRF to learn a GENERAL model that can learn common features from multiple scenes and then generalize to new scenes.
>
>
>  In stark contrast, being also built on the successful volume rendering of NeRF, our GRF is a intuitively simple yet valid method to discover the general and shareable pixel-level patterns for multi-scene/novel-scene representation and reconstruction. This is key to the future of machine intelligence. Our code will be made available on GitHub for reproducing all experimental results.
>  We believe this would advance the field of 3D scene understanding and have true impact for the community.
>
>
> It might be helpful to note that the GENERATIVE model (GRAF) is completely different from the GENERAL model (GRF) in terms of the final objective and the design principle. To avoid the confusion of names, we are open to change a new name if it is suggested to do so.
>
> Overall, we highly appreciate all efforts of the reviewer to help us improve the manuscript. It would be great if the reviewer could reconsider the rating based on our latest updates.

---

### Official Review · AnonReviewer4 · 2020-10-29
**This paper proposes a neural rendering model named General Radiance Field (GRF) to achieve 3D representation learning and novel view synthesis. Different from the previous neural rendering methods NeRF and SRN that directly learn the mapping from pose information to color, GRF learns to aggregate the 2D features of different observations via attention mechanism given a 3D location and view pose.**

**Rating:** 6
**Confidence:** 4

**Review:**

Pros:
+ This paper proposes a method for synthesizing 3D scenes in novel view by treating the network as a general radiance field. It seems that the method is more like traditional manner based on multiple view geometry. It enables the generalization ability of rendering unseen test data in contrast to the most related work, NeRF (Mildenhall et al., 2020).
+ The proposed GRF can empirically generalize to novel scenes. Experimental results show that the proposed method achieves better performance on several large-scale datasets.
+  In general, this paper is well-written and easy to read.

Concerns:
- The novelty is limited as the concept of this paper is highly similar to NeRF, despite a significant improvement is the general feature for novel view rendering. Yet I have few questions about general features for 3D points.
1.  As depicted, the aggregator that assembles the 2D features of each 3D point from multiple views handles the visual occlusion implicitly via an attention process without depth scans. How does it work on unseen scenes? The network may have no knowledge about the structure of the novel scenes. Are there any failure examples of such cases?
2.  Rendering a query 3D point p requires the (r_p, g_p, b_p) and the density d_p. As described in Eq. (6), the color channels are estimated through MLPs with learned feature \bar{F}_p and the query viewpoint \mathcal{V}_p as input. Nonetheless, the density function in Eq. (5) does not require the query viewpoint as input. I think the density of 3D points for volume rendering should be dependent on the viewing ray as well. How to explain the difference between Eq. (5) and Eq. (6)?

- The evaluation of generalization for the novel scene of the proposed model is still not convincing enough, which needs improvements.
1.  In section 4.1, the performance of SRN is better than the proposed methods in almost all situations. The authors give an explanation that SRN requires to be retrained on novel scenes to optimize the latent code. I suggest that the authors should evaluate the performance of SRN on novel scenes without retraining process to validate the argument.
2.  The experiments only conduct on the novel scene of the same category which share the similar features. I consider that SRN can also handle this situation by the learned latent code and hypernetwork. The authors are suggested to train the proposed GRF on a large amount of data with different categories of objects to learn the general feature for attention mechanism, and evaluate the model on the new object to validate the generalization.

---

> ### Author Response · Authors · 2020-11-24
> **Response to Reviewer 4**
>
> **Concern 1**\
> \
> As depicted, the aggregator that assembles the 2D features of each
> 3D point from multiple views handles the visual occlusion implicitly via
> an attention process without depth scans. How does it work on unseen
> scenes? The network may have no knowledge about the structure of the
> novel scenes. Are there any failure examples of such cases?\
> \
> **Response**:\
> This is a fundamental question. Given sparse views of an unseen scene,
> the CNN module of our trained GRF can extract hierarchical pixel
> features including the pixel local patterns and the possible larger
> shape information. These features are usually believed to be common and
> shared across different objects and scenes. This allows our model to
> have a certain level of generalization capability across unseen scenes.
>
> Basically, the attention mechanism is designed to select the most
> important pixel features among many for rendering a novel pixel. This
> module is likely to assign higher attention scores to visually similar
> and visible pixel patches.
>
> In order to evaluate the effectiveness of the attention module, we
> conduct an experiment to visualize and analyze the learned attention
> scores on novel objects. Details are presented in the appendix (A.4). It
> shows that the attention module can truly drive the network to focus on
> the visible pixel local features among the multiple intersected light
> rays. This result is also added into the section Ablation Study.
> ------------------------------------------------------------------------
> **Concern 2**\
> Rendering a query 3D point p requires the (r_p, g_p, b_p) and the
> density d_p. As described in Eq. (6), the color channels are estimated
> through MLPs with learned feature $\bar{F}_p$ and the query viewpoint
> $\mathcal{V}_p$ as input. Nonetheless, the density function in Eq. (5)
> does not require the query viewpoint as input. I think the density of 3D
> points for volume rendering should be dependent on the viewing ray as
> well. How to explain the difference between Eq. (5) and Eq. (6)?
>
> **Response**:\
> Here is the clarification. The volumetric density $d_p$ is modeled as a
> function of the point features $\mathbf{\bar{F}}_p$ only, because this
> allows the estimated density $d_p$ at the query point $p$ to remain
> unchanged under different query viewpoints, thus maintaining consistent
> geometry for each point. We update the paper accordingly in section 3.5,
> the last paragraph of page 5.
> ------------------------------------------------------------------------
> **Concern 3**\
> In section 4.1, the performance of SRN is better than the proposed
> methods in almost all situations. The authors give an explanation that
> SRN requires to be retrained on novel scenes to optimize the latent
> code. I suggest that the authors should evaluate the performance of SRN
> on novel scenes without retraining process to validate the argument.\
>
> **Response**:\
> Thanks for the valuable suggestion. In the revised paper, we conducted
> additional experiments as suggested in section 4.1, the last paragraph
> of page 6. The qualitative results are presented in Figure 4, clearly
> demonstrating the advantage of our GRF over SRNs when directly being
> tested on novel objects.
> ------------------------------------------------------------------------
> **Concern 4**\
> The evaluation of generalization for the novel scene of the proposed
> model is still not convincing enough, which needs improvements.
>
> The experiments only conduct on the novel scene of the same category
> which share the similar features. I consider that SRN can also handle
> this situation by the learned latent code and hypernetwork. The authors
> are suggested to train the proposed GRF on a large amount of data with
> different categories of objects to learn the general feature for
> attention mechanism, and evaluate the model on the new object to
> validate the generalization.
>
> **Response**:\
> We agree with the reviewer that our GRF can be trained on a large amount
> of objects belonging to different categories. However, we empirically
> find that such experiments would take up to a few months to finish on a
> modern GPU. Alternatively, we conduct additional experiments on the
> Synthetic-NeRF dataset to evaluate the generalization across novel
> scenes. In particular, our GRF is trained on four different scenes, and
> then directly tested on the remaining completely different four scenes.
> The results show that our GRF can indeed generalize well across novel
> scenes. The details are presented in the revised paper, the section 4.2
> page 7 and 8.
> ------------------------------------------------------------------------

---

### Official Review · AnonReviewer3 · 2020-10-29
**Accept**

**Rating:** 7
**Confidence:** 4

**Review:**

This paper presents an extension of NeRF. The key idea is to represent a 3D
scene as a collection of K "posed" images. A network is trained to take these
images, viewing information and a query point as input and output density (soft
occupancy) and RGB reflectance color. In contrast to NeRF, this methods learns a
a mapping from the input images to feature vectors and thus promises to
generalize across inputs (whereas NeRF uses weight-encoding). On a detailed
level, this method also uses multiview consistency during training, though there
is not an ablation I could find that demonstrates the effectiveness of this
delta.

I recommend accepting this paper to ICLR. I like the idea of using the K input
images as parameters controling the scene. My main criticisms are in the
exposition of the paper and the experiments. I hope the exposition can be
improved in revisions and am comfortable leaving the experimental shortcomings
to future work.

My main source of confusion reading this paper centers around the input
"viewpoint". The following comments are made in order of the paper's exposition
and hopefully highlight the path of my confusion:

Please define what's meant by "posed 2D images".

Are the viewpoints V1, ... Vk corresponding to the K input images really just
the 3D position of the camera focal point? Or does this also include the view
direction and camera intrinsics? If it does include other information then
perhaps use a different symbol for the query viewpoint.

Near Figure 2, the discussion of "viewpoint" for the image is really confusing.
Do all pixels of an input share the same viewpoint position (seems to be the
case. This seems awkward, but best matches the written explanation.

Or is the image "placed" into the 3D scene according to a pinhole camera's
transformation matrix, so that each pixel gets a unique 3d position associated
to it? This seems more appropriate, but would not match the text or explanation.

The reprojection step appears to assume access to more than the camera center.

Much later the paper says Vk "including extrinsics and intrinsics". This is a
fairly abusive notation. This should be clarified early on and the confusion
issue with the image augmentation with viewpoint position remains.

If the intro/abstract made it clear that the input is K images and full camera
information and a different symbol was used for camera info and query viewpoint,
much of this confusion would go away.

In the motivation of this paper, there is special emphasis on representing
"scenes", yet the experiments are on "single objects", just as other methods are
criticized for being limited to. Can the proposed method be used to represented
a full indoor room (not just a 360° video; but view from anywhere in any
direction)?

The examples in the appendix are more like photos+depth type examples than what
I would consider a full "scene". Perhaps tone down/clarify what is claimed in
the introduction.

I would also like to get a sense of how stable / gracefully degrading this
method is to the K input images. For example, if I only input images of the
front of an object, what will happen when viewing the back? What if most of the
images are from one direction, will this bias affect views elsewhere?

Does the method generalize to scenarios with ≠ K input images? I suppose one
could use the zero vectors trick for the <K images, but I wonder about
degradation. What about >K?

It is not really fair to write that "mose methods ... require ground truth 3D
geometry for supervision". NERF/IDR/SIREN and when considering noisy point
clouds SAL/SAL++ do not need ground truth 3D geometry.

The paper writes "This simple design of GRF follows the principle of classic
multi-view geometry (Hartley & Zisserman, 2004), therefore guaranteeing the
learned implicit representations meaningful and multi-view consistent." I do not
see how this setup provides any formal guarantee. In the appendix it appears
that following NeRF this paper predicts the density from the query position and
not the viewing direction. Is that simply inherited here? The multiview aspects
during training simply harmonize with this choice but don't provide any
guarantee that I can understand.

"for very query" --> "for every query"

---

> ### Author Response · Authors · 2020-11-24
> **Response to Reviewer 3**
>
> **Concern 1**\
> Notational inconsistency and clarification relating to 'viewpoint' and 'posed images'\
> \
> **Response**:\
> We thank the reviewer for pointing out the inconsistency of "viewpoint"
> and the symbols. In the revised paper, we made the following
> modifications: 1) The "viewpoint" is clearly defined as the camera
> location xyz. 2) All " posed 2D images" are replaced by "2D images with
> camera poses and intrinsics". 3) Near Figure 2, the discussion about
> viewpoint is updated with clear specifications. 4) Equation 1, Figure 1
> and the symbols are all updated with clear definition. The whole paper
> now has consistent meanings for the viewpoint.
>
>
> ------------------------------------------------------------------------
>
> **Concern 2**\
> In the motivation of this paper, there is special emphasis on
> representing \"scenes\", yet the experiments are on \"single objects\",
> just as other methods are criticized for being limited to. Can the
> proposed method be used to represented a full indoor room (not just a
> 360° video; but view from anywhere in any direction)?
>
> **Response**:\
> This is a good point. In principle, our GRF formulates the 3D structure
> via per 3D query point, which is agnostic to the complexity of scenes.
> The success on both Synthetic-NeRF and the real-world datasets also
> demonstrates that our GRF can indeed represent a certain level of
> complex 3D scenes. However, it is still challenging to directly evaluate
> on large-scale 3D scenes such as full indoor rooms or extreme large
> outdoor space. In order to avoid the confusion, we made the following
> changes in the revised paper: 1) In Introduction, the first paragraph of
> page 2, we specify \"the complex 3D scenes\" as \"multiple objects with
> cluttered background\". 2) In Section 4.3, the last paragraph of page 8,
> we analyse the difficulties of recovering large-scale 3D scenes and it
> is left for future exploration.
>
> ------------------------------------------------------------------------
>
> **Concern 3**\
> I would also like to get a sense of how stable / gracefully degrading
> this method is to the K input images. For example, if I only input
> images of the front of an object, what will happen when viewing the
> back? What if most of the images are from one direction, will this bias
> affect views elsewhere?\
> \
> **Response**:\
> This is an interesting question. We conduct an experiment accordingly
> and the results are presented in the appendix (A.5). 1) In the extreme
> case, i.e., 1-view reconstruction, our GRF is still able to recover the
> general 3D shape of the unseen object, including the visually occluded
> parts, primarily because our CNN model learns the hierarchical features
> including the high-level shapes. 2) Given more input views, the
> originally occluded parts tend to be observed from some viewing angles,
> and then these parts can be reconstructed better and better. This shows
> that our GRF is indeed able to effectively identify the corresponding
> useful pixel features for more accurately recovering shape and
> appearance.
>
> ------------------------------------------------------------------------
>
> **Concern 4**\
> Does the method generalize to scenarios with $\neq$ K input images?\
> \
> **Response**:\
> Since the used attention modules (AttSets or Slot-Att) are able to
> aggregate an arbitrary number of feature vectors, our GRF can naturally
> take any number of input images without needing the zero vector trick.
> We have conducted four groups of experiments for K = (1,2,5,10) in
> testing, where the model is trained with 5 images per object. Details
> are shown in the appendix (A.5). The results show that with less input
> images, the quality of shapes indeed degrades, especially the visually
> occluded parts.
>
> ------------------------------------------------------------------------
>
> **Concern 5**\
> It is not really fair to write that \"mose methods \... require ground
> truth 3D geometry for supervision\". NERF/IDR/SIREN and when considering
> noisy point clouds SAL/SAL++ do not need ground truth 3D geometry.\
> \
> **Response**:\
> We thank the reviewer for pointing out the related work. In the revised
> paper, we rephrased the sentence and briefly discussed SAL/SAL++ in the
> section Related Work.
>
> ------------------------------------------------------------------------
> **Concern 6**\
> The paper writes \"This simple design of GRF follows the principle of
> classic multi-view geometry (Hartley & Zisserman, 2004), therefore
> guaranteeing the learned implicit representations meaningful and
> multi-view consistent.\" I do not see how this setup provides any formal
> guarantee.
>
> \"for very query\" -- $>$ \"for every query\"\
> \
> **Response**:\
> We agree that the word "guaranteeing" is not appropriate. It is replaced
> by more suitable words such as "empirically remaining" and \"leading
> to\" throughout the paper to avoid the confusion. In addition, we
> further discuss the reasons in section 3.1, the last paragraph of page
> 3. Typos are corrected.

---

> > ### Comment · AnonReviewer3 · 2020-11-24
> > **continue to advocate for acceptance**
> >
> > Thank you for the detailed responses and changes. These changes improve this paper further.
> >
> > With these changes and having read the other reviews and responses, I'm feeling more confident in my review of "7: Good paper, accept". I advocate for acceptance.

---

> > > ### Author Response · Authors · 2020-11-25
> > > **Response to Reviewer's Advocation**
> > >
> > > We thank the reviewer for the support of our paper, and hope your valuable comments can be helpful for the audience and peer reviewers to accurately evaluate our contributions to the community.

---

### Decision · Program_Chairs · 2021-01-07
**Final Decision**

**Decision:**

Reject

**Comment:**

The paper presents an extension of recent implicit representations for view synthesis, such as NeRF. The presented formulation accepts an image set as input at test time, and can thus in principle be applied to new scenes. The idea is sound, but reviewers had concerns with the presentation and the experimental results. The work is primarily evaluated on the simplistic ShapeNet domain, which a number of reviewers found unconvincing. Concerns remain even after the authors' responses, and the AC agrees that the work can benefit from further investment before it is published.

---

> ### Comment · ~Bo_Yang7 · 2021-08-16
> **Accepted by ICCV 2021**
>
> Thanks for the feedback. The updated version has been published at ICCV 2021. Welcome to check out our paper and code.
>
> Full paper: https://arxiv.org/abs/2010.04595
>
> Code: https://github.com/alextrevithick/GRF